



# Self-enhanced aerosol–fog interactions in two successive radiation fog events in the Yangtze River Delta, China: A simulation study

Naifu Shao[1], Chunsong Lu[1], Xingcan Jia[2], Yuan Wang[3], Yubin Li[1], Yan Yin[1], Bin Zhu[1], Tianliang Zhao[1], Duanyang Liu[4,5], Shengjie Niu[1,6], Shuxiang Fan[1], Shuqi Yan[4,5], Jingjing Lv[1]

[1]Key Laboratory for Aerosol-Cloud-Precipitation of China Meteorological Administration/Collaborative Innovation Centre on Forecast and Evaluation of Meteorological Disasters (CIC-FEMD), Nanjing University of Information Science & Technology, Nanjing, China
[2]Institute of Urban Meteorology, China Meteorological Administration (CMA), Beijing 100089, China
[3]Collaborative Innovation Centre for Western Ecological Safety, Lanzhou University, Lanzhou 730000, China.
[4]Nanjing Joint Institute for Atmospheric Sciences, Nanjing 211112, China
[5]Key Laboratory of Transportation Meteorology, CMA, Nanjing 210009, China
[6]College of Safety Science and Engineering, Nanjing Technology University, Nanjing 210009, China

*Correspondence to*: Chunsong Lu (luchunsong110@gmail.com)

**Abstract.** Aerosol–fog interactions (AFIs) play pivotal roles in the fog cycle. However, few studies have focused on the differences in AFIs between two successive radiation fog events and the underlying mechanisms. To fill this knowledge gap, our study simulates two successive radiation fog events in the Yangtze River Delta, China, using the Weather Research and Forecasting model coupled with Chemistry (WRF-Chem). Our simulations indicate that AFIs in the first fog (Fog1) promote AFIs in the second one (Fog2), resulting in higher number concentration, smaller droplet size, larger fog optical depth, wider fog distribution, and longer fog lifetime in Fog2 than in Fog1. This phenomenon is defined as the self-enhanced AFIs, which are related to the following physical factors. The first one is conducive meteorological conditions between the two fog events, including low temperature, high humidity and high stability. The second one is the feedbacks between microphysics and radiative cooling. A higher fog droplet number concentration increases the liquid water path and fog optical depth, thereby enhancing the long-wave radiative cooling and condensation near the fog top. The third one is





the feedbacks between macrophysics, radiation, and turbulence. A higher fog top presents stronger long-wave radiative cooling near the fog top than near the fog base, which weakens

temperature inversion and strengthens turbulence, ultimately increasing the fog-top height and fog area. In summary, AFIs postpone the dissipation of Fog1 due to these two feedbacks and generate more conducive meteorological conditions before Fog2 than before Fog1. These more conducive conditions promote the earlier formation of Fog2, further enhancing the two feedbacks and strengthening the AFIs. Our findings are critical for studying AFIs and shed new

light on aerosol–cloud interactions.

## 1 Introduction

Fog comprises many water droplets or ice crystals suspended above the ground (WMO, 1992). This leads to environmental pollution and low visibility, affecting the human health, transportation, and power system (Niu et al., 2010). There exist uncertainties in fog forecasting

(Zhou and Du, 2010; Zhou et al., 2011). An important reason is that the physical processes of fog remain unclear, which impedes the related parameterisation. To better understand the physical processes of fog, comprehensive studies have been conducted based on observations and simulations  (Fernando et al., 2021; Gultepe et al., 2014; Guo et al., 2015; Hammer et al., 2014; Liu et al., 2011; Price et al., 2018; Shen et al., 2018; Wang et al., 2021). The pivotal roles

of aerosols and the planetary boundary layer (PBL) in these processes have been proven (Boutle et al., 2018; Niu et al., 2011; Quan et al., 2021).

  Fog is a type of cloud suspended near the surface (Kim and Yum, 2010, 2013). Studies on aerosol–cloud interactions revealed that increasing aerosol loading increased cloud droplet concentration, thereby increasing the cloud optical depth under a constant liquid water content

(LWC) (Garrett and Zhao, 2006; Twomey, 1977; Wang et al., 2013; Wang et al., 2018; Zhao and Garrett, 2015). Guo et al. (2021) showed that the radiation effects between fog and cloud



were similar. In different emission backgrounds, Quan et al. (2011) found that the fog number concentration ($N_f$) was larger than 1,000 cm$^{-3}$ and effective radius ($R_e$) was approximately 7 μm under polluted condition. Whereas, Wang et al. (2021) showed that $N_f$ was smaller than 100 cm$^{-3}$ and $R_e$ was approximately 9 μm under clean condition. Several simulation studies reproduced these observations and demonstrated the complex impacts of aerosol–fog interactions (AFIs) on fog micro- and macrophysics (Jia et al., 2019; Maalick et al., 2016; Stolaki et al., 2015; Yan et al., 2020). Regarding fog microphysics, increasing aerosol loading increased $N_f$ and LWC but decreased $R_e$ due to increased activation and condensation in simulations (Jia et al., 2019; Stolaki et al., 2015; Yan et al., 2020). Regarding fog macrophysics, some model studies revealed that increased aerosol loading increased the fog-top height (Jia et al., 2019; Stolaki et al., 2015) and prolonged the fog lifetime by delaying its dissipation (Quan et al., 2021; Yan et al., 2021).

Furthermore, previous studies found that meteorological conditions played crucial roles in aerosol–cloud interactions as well as cloud macro- and microphysics (Ackerman et al., 2004; Kumar et al., 2017; Kumar et al., 2021; Liu et al., 2019; Liu et al., 2020; Toll et al., 2019). Similarly, studies on fog showed that AFIs were affected by meteorological conditions in the PBL (e.g., radiation, thermodynamics, and dynamics), which further affected fog micro- and macrophysics (Haeffelin et al., 2010). Liu et al. (2010), Kim and Yum (2011), and Kim and Yum (2012) noted that radiative cooling was an important factor for temperature inversion, providing stable conditions for fog formation. According to Zhou and Ferrier (2008), turbulence may suppress or deepen the fog-top height, which was related to the critical turbulence coefficient. If temperature inversion was weak, excessive vertical turbulent mixing delayed fog formation (Maronga and Bosveld, 2017). However, if temperature inversion was sufficiently strong, vertical turbulent mixing at the middle and fog base increased the fog top height, as proposed based on observations (Ye et al., 2015) and simulations (Porson et al., 2011).





Consequently, turbulence may affect fog macrophysics. Furthermore, aerosols affect turbulence, thereby impacting fog macrophysics (Jia et al., 2019; Quan et al., 2021). A previous qualitative analysis revealed that aerosols promoted turbulence and horizontal distribution because of

weaker temperature inversion (Jia et al., 2019).

It is noteworthy that the understanding of AFIs remains limited (Poku et al., 2021; Schwenkel and Maronga, 2019). In particular, the evolution of AFIs in successive fog events remains unknown, because many previous studies mainly focused on a single fog event or analysed multiple fog events as a whole. Additionally, the evolution of AFIs is helpful to study the

evolution of aerosol–cloud interactions. Therefore, to improve our understanding of AFIs, two successive radiation fog events in the Yangtze River Delta (YRD) region in China are simulated in this paper using the Weather Research and Forecasting model coupled with Chemistry (WRF-Chem). Our specific objectives are twofold. First, we seek to answer the following questions: which fog scenario is stronger and experiences stronger AFIs? Can AFIs become

self-enhanced? Second, the two fog scenarios provide an excellent opportunity to analyse AFIs as a chain, i.e., how aerosol affects the first fog scenario, how the first fog scenario affects radiation and the PBL structure, and then how radiation and the PBL affect AFIs in the second fog scenario.

The rest of the article is organised as follows. Section 2 presents descriptions of the two

successive fog events, experimental design, and data source. Section 3 presents simulation verification. Section 4 shows that AFIs in the second fog event are stronger than those in the first one. Section 5 presents the physical mechanisms underlying self-enhanced AFIs. Section 6 summarises the conclusions.



## 2   Experimental design and data source

Here, we study AFIs with two successive radiation fog events in the YRD region. There is massive aerosol loading in the YRD due to anthropogenic emissions (Ding et al., 2016; Shi et al., 2008; Wang et al., 2019; Yan et al., 2019). On 26–27 November 2018, two successive radiation fog events occurred in northern YRD. The first fog event is called Fog1, and the second one is called Fog2. Ground-based observations at the Nanjing site (32.2°N 118.7°E)

show that the two fog events (visibility <1,000 m) occurred with high relative humidity, low temperature, and weak wind speed (Fig. 1). As shown in Fig. S1, the surface is controlled by a high–pressure system with cold and moist air in the northern YRD at 20:00 local standard time (LST) (LST = Universal Time Coordinated + 8 h) on 26 and 27 November 2018. WRF-Chem (version 4.1.3) is employed to simulate the two successive radiation fog events. WRF-Chem

couples physical and chemical processes; therefore, it has been widely utilised to study AFIs (Jia et al., 2019; Lee et al., 2016; Yan et al., 2020; Yan et al., 2021). The model is integrated from 14:00 LST on 24 November 2018 to 14:00 LST on 27 November 2018, with the first 24 hours regarded as the spin-up time. As shown in Fig. S2, the model is configured using three nested domains, and the domain centres are all located in Nanjing. The three nested domains

are $90 \times 122$, $118 \times 142$, and $130 \times 154$ grid cells with resolutions of 27, 9, and 3 km, respectively. The simulation area covers the major weather system, which can affect the YRD. There are 36 vertical levels in the model, of which 17 are located in the lowest 500 m above the ground. Moreover, Yang et al. (2019) noted better fog simulation performance when the bottom layer was 8 m above the ground, because this layer affected the interactions between fog and

surface flux. Consequently, we set the model bottom layer as 8 m in the present study. In addition, the model is driven by the National Centre for Environmental Prediction (NCEP) Final (FNL) 1°×1° reanalysis data (https://rda.ucar.edu/datasets/ds083.2/) (Ding et al., 2019; Jia et al., 2019). The Multiresolution Emission Inventory for China (MEIC) database



([http://meicmodel.org](http://meicmodel.org)) is used for anthropogenic emissions in the model (Li et al., 2017a; Zheng
et al., 2018).

Table 1 shows the parameterisation schemes of physical processes used in the present
study. The microphysics scheme is Morrison (Morrison et al., 2005) coupled with the activation
scheme  (Abdul-Razzak, 2002). The PBL scheme is MYNN2.5 (Nakanishi and Niino, 2009).
The radiation schemes are coupled with aerosol–cloud–radiation interactions. The long- and
short-wave radiation schemes are RRTMG (Iacono et al., 2008) and Goddard (Matsui et al.,
2020), respectively. The cumulus scheme is Grell 3D (Grell and Dévényi, 2002). The chemistry
schemes are MOSAIC-4 bins (Zaveri et al., 2008) and CBMZ (Zaveri and Peters, 1999).

For model verification, meteorological data are retrieved from the China Meteorological
Administration (http://www.nmic.cn/), satellite data are retrieved from the Himawari-8
geostationary satellite (https://www. eorc.jaxa.jp/ptree/index.html), and $PM_{2.5}$ mass
concentration data are from the Ministry of Environmental Protection (https://quotsoft.net/air/).
Grids satisfying the following three criteria are recognised as simulated foggy grids (Jia et al.,
2019; Zhao et al., 2013): fog water mixing ratio exceeding 0.01g $kg^{-1}$, $N_f > 1$ $cm^{-3}$, and fog base
touching the ground.

To investigate the effects of AFIs on the fog macro- and microphysics, a sensitivity
experiment is conducted. In the control run (i.e. polluted condition), emission intensity is
adopted directly from the MEIC database. In the sensitivity run, emission intensity is multiplied
by 0.05 as the clean condition (Jia et al., 2019; Yan et al., 2021).

## 3    Simulation verification

Simulation verifications for temperature, relative humidity, and wind speed are shown in Fig.
2. The correlation coefficients of 2 m temperature ($T_{2m}$), 2 m relative humidity ($RH_{2m}$), and 10




m wind speed (WS$_{10m}$) between the simulations and observations are 0.9, 0.9, and 0.6, respectively, passing the significance test at 99%. Therefore, the simulations are generally consistent with the observations. The mean deviations of T$_{2m}$, RH$_{2m}$, and WS$_{10m}$ between the

simulations and observations are 1.0 °C, 2.7%, and 0.4 m s$^{-1}$, respectively, consistent with evaluation results in studies by Hu et al. (2021), Gao et al. (2016), and Yang et al. (2022). Figure 3 shows the evaluation of PM$_{2.5}$ distribution, and Table 2 summarises statistics of mean mass concentration of PM$_{2.5}$ based on the method proposed by Boylan and Russell (2006). The normalised mean bias (NMB), normalised mean error (NME), mean fractional bias (MFB), and

mean fractional error (MFE) between the simulations and observations are 25%, 30%, 24%, and 28%, respectively (Eqs. S1-S4). Although the PM$_{2.5}$ mass concentration is overestimated, it remains within a reasonable range (Shu et al., 2021; Yang et al., 2022; Zhai et al., 2018).

Figure 4 shows the evaluation of fog spatial distribution. The simulated liquid water path (LWP) distribution is compared with the Himawari-8 visible cloud images and ground-based

observations (red points in Fig. 4) at 08:00 LST on 26 and 27 November 2018. The simulated spatial distribution of fog is consistent with satellite and ground-based observations. Furthermore, the Heidke skill score (HSS) is used to evaluate the simulations (Barnston, 1992):

$$HSS = \frac{2(ad-bc)}{(a+c)(c+d)+(a+b)(b+d)} \tag{1}$$

where elements $a$–$d$ are the numbers of "hits", "false alarms", "misses", and "correct negatives",

respectively. The HSS are 0.34 and 0.36 in Fog1 and Fog2, respectively, indicating values close to previous reports (Mecikalski et al., 2008; Xu et al., 2020; Yamane et al., 2010). Therefore, the model can generally capture the fog spatial distribution.





## 4 Aerosol–fog interactions in the second fog event are stronger than those in the first one

Here, we analyse the fog macro- and microphysical characteristics under the clean and polluted conditions (Fig. 5). To ensure sufficient sample size for statistical analysis, only data with the fog area fraction larger than 5% are analysed. The fog area fraction is calculated as the number of foggy grid cells divided by the total number of grids in domain 03.

Differences between the clean and polluted conditions reveal that AFIs affect fog (Fig. 5a-b) macro- and microphysics. Compared to fog microphysics under clean conditions, $N_f$ and LWC in Fog1 increase by respectively 463.0% and 81.7% but $R_e$ decreased by 32.1% under polluted conditions. Furthermore, because of the AFIs, $N_f$ and LWC in Fog2 increase by respectively 672.4% and 113.5% but $R_e$ decreases by 40.0%. Thus, AFIs in Fog2 are stronger than those in Fog1 in terms of microphysics (Fig. 5c). Similarly, the effects of AFIs on fog macrophysics are stronger in Fog2. Compared with values under clean conditions, the fog area, fog-top height, and duration in Fog1 increase by respectively 23.1%, 109.6%, and 20.0% under polluted conditions; the corresponding values in Fog2 are larger (34.9%, 350.5%, and 25.0%, respectively). In addition, LWP and FOD show similar trends. Figure 5c further confirms the above conclusions based on direct comparison between Fog2 and Fog1.

Figure 5d presents the fog duration affected by AFIs. Fog duration is determined by the time of fog formation and dissipation. Fog duration is primarily extended because of the delaying of fog dissipation by aerosols, as reported previously (Jia et al., 2019; Quan et al., 2021). In this paper, aerosols not only postpone fog dissipation but also promoted earlier fog formation, particularly during Fog2 (Fig. 5d). To investigate the aerosol effect on the Fog2 formation stage, fog spatial distribution at the formation stage from 19:00 LST to 21:00 LST on 26 November is examined, as shown in Figure 6. The fog area is rather small at 19:00 LST under both polluted and clean conditions. At 20:00 LST, in grid cells located outside the black





box, fog formation is similar under both polluted and clean conditions. Inside the black box,
there are several foggy grid cells under polluted conditions. At 21:00 LST, fog area in the black
box further expands under polluted conditions. However, there is almost no fog in the black
box at 20:00 LST and 21:00 LST under clean conditions. Therefore, aerosols promote earlier
formation of Fog2, which is primarily caused by meteorological conditions in the PBL inside
the black box. In addition, the fog area outside the black box is larger under polluted conditions
than under clean conditions, which is mainly related to the stronger turbulence diffusion under
polluted condition. Detailed analysis is described in Sect. 5.

Further, to quantitatively evaluate the strength of AFIs in the two fog events, we examine
the responses of fog optical depth (FOD) to changes in $N_f$ (Eq. 2) (Ghan et al., 2016):

$$\frac{\Delta \ln FOD}{\Delta \ln N_f} = \frac{\Delta \ln LWP}{\Delta \ln N_f} - \frac{\Delta \ln R_e}{\Delta \ln N_f} \tag{2}$$

As shown in Table 3, the strength of AFIs in Fog2 (1.32) is larger than that in Fog1 (0.98),
and the contribution from $\Delta \ln LWP/\Delta \ln N_f$ (0.76) is larger than that from $-\Delta \ln R_e/\Delta \ln N_f$ (0.22).
Results for Fog2 are similar. Relative changes in the above properties between Fog1 and Fog2
are calculated as (Fog2 − Fog1)/Fog1. The values of $\Delta \ln FOD/\Delta \ln N_f$, $\Delta \ln LWP/\Delta \ln N_f$, as well
as $-\Delta \ln R_e/\Delta \ln N_f$ are 34.7%, 42.1%, and 9.1% larger in Fog2 than in Fog1, respectively. These
numbers quantitatively confirm self-enhancing AFIs and indicate that LWP is the dominant
factor for enhancing AFIs. LWP depends on the fog-top height and LWC. As shown in Fig. 5a–
b, when aerosol loading changes from clean to pollution, the rate of increase in fog-top height
in Fog2 (350.5%) is much larger than that in Fog1 (109.6%). Although the increase of LWC in
Fog2 (113.5%) is also larger than that in Fog1 (81.7%), the magnitude of increase in LWC is
smaller than that increase in fog-top height, indicating that AFIs are more sensitive to fog-top
height than to LWC.





## 5 Physical mechanisms underlying self-enhanced aerosol–fog interactions

### 5.1. More conducive meteorological conditions before Fog2

Meteorological conditions in the PBL affect the fog formation time and AFIs during fog events. As shown in Table 4, under clean conditions, $RH_{2m}$ before Fog2 formation is higher and PBL height (PBLH) is lower than those before Fog1 formation in domain 03. Furthermore, before Fog2 formation, relative humidity is higher and lower PBLH is lower under polluted conditions than under clean conditions. Therefore, aerosols generate more conducive meteorological conditions for Fog2 formation during two successive fog events.

To further analyse how aerosols promote Fog2 formation, we focus on the black box in Fig. 6, as described in Sect. 4 and by Yan et al. (2021). The regional average differences in the total optical depth (TOD), downwelling short-wave radiation (SW) at the ground, $T_{2m}$, PBLH, $RH_{2m}$, and water vapour mixing ratio ($Qv_{bot}$) at the model bottom layer (8 m) in the black box between polluted and clean conditions are calculated (Fig. 7). During the daytime before Fog2 formation, meteorological conditions in the PBL are affected by AFIs at the Fog1 dissipation stage. A larger TOD induced by AFIs leads to lower SW, $T_{2m}$, and PBLH. Notably, $Qv_{bot}$ under polluted conditions is lower than that under clean conditions before complete dissipation of Fog1, because of less fog water evaporation. When the fog dissipates completely, the lower PBLH accumulates more water vapour, increasing $Qv_{bot}$ and $RH_{2m}$. The positive feedbacks between AFIs and PBL are similar to the feedbacks between aerosols and PBL reviewed by Li et al. (2017b). Further, the feedback mechanism between aerosol and PBL introduced by Zhong et al. (2018) supports the daytime feedbacks between AFIs and PBL in the present study. Additionally, although aerosol extinction should be considered in TOD, Yan et al. (2021) suggested that AFIs produce a more remarkable impact on PBL than aerosol–radiation interactions. In this paper, we show that the largest temperature difference appear during the



fog dissipation stage due to AFIs (Fig. 7). Therefore, lower temperature, higher relative humidity, and stronger stability result from AFIs in Fog1, contributing to the earlier formation of Fog2.

AFIs result in lower temperature, higher relative humidity, and stronger stability by affecting solar radiation during the daytime. How can these conducive conditions be maintained
after the sunset around 17:00 LST? Figure 8a shows that cold advection is the major reason responsible for the difference in temperature between polluted and clean conditions. We further seek to unveil the reason cold advection is stronger under polluted conditions. Figure 8b shows a cold centre, with wind diverging outward from it. The cold centre is related to lower temperature under polluted conditions due to AFIs in Fog1. Likewise, Steeneveld and De Bode
(2018) pointed out that fog appeared earlier with cold advection. In addition, lower PBLH induced by aerosols promotes the maintenance of higher humidity and stronger stability.

Overall, as mentioned above, the more conducive meteorological conditions promote Fog2 formation due to AFIs at the Fog1 dissipation stage. Furthermore, these interactions enhance the feedbacks in the fog physical processes, thus rendering AFIs self-enhanced. Details are
discussed in Sect. 5.2 and 5.3.

## 5.2. Feedbacks between microphysics and long-wave cooling

Section 5.1 reveals the mechanism through which AFIs in Fog1 lead to more conducive meteorological conditions before Fog2 formation. In Sect. 5.2, we demonstrate how conducive meteorological conditions play fundamental roles in promoting the feedbacks between
microphysics and long-wave cooling, resulting in self-enhanced AFIs.

As shown in Fig. 5c, LWC and $N_f$ in Fog2 are larger than those in Fog1 because lower temperature and higher humidity are more conducive for aerosol activation and fog condensation (Petters and Kreidenweis, 2007; Simmel and Wurzler, 2006). Due to competition





for available water vapour (Mazoyer et al., 2022; Yum and Hudson, 2005), $R_e$ in Fog2 is smaller

than that in Fog1. Consequently, FOD in Fog2 is larger than that in Fog1. Additionally, increased FOD in Fog2 triggers stronger positive feedbacks between microphysics and long-wave cooling, further enhancing cooling, activation, and condensation and thereby increasing $N_f$ and LWC. Jia et al. (2019) emphasised that aerosols promoted these positive feedbacks. The present study further highlights the synergistic effects of aerosols and meteorological conditions

on the enhancement of positive feedbacks, which promote AFIs in Fog2.

To better understand how the above positive feedbacks affect AFIs, Fig. 9 presents the FOD per unit height (FOD/$\Delta$h), radiative cooling rate ($T_{LW}$), condensational growth rate (LWC$_{COND}$), and LWC tendency due to vertical mixing (LWC$_{mixing}$) in the two successive fog events. Radiative cooling is the strongest near the fog top and weakest at the fog base (Ducongé

et al., 2020; Mazoyer et al., 2017; Wærsted et al., 2017). Consequently, LWC$_{COND}$ and LWC$_{mixing}$ both follow similar profiles in response to radiative cooling. Therefore, if the vertical profiles of the three terms use absolute height, they will be distorted. To overcome this problem, physical quantities are normalised by the fog-top height. Compared with those in Fog1, larger FOD (Fig. 9a-b), stronger long-wave radiative cooling (Fig. 9c-d), and more condensation (Fig.

9e-f) near the fog top are noted in Fog2, which further increases LWC and fog-top height in Fog2 (black and purple lines). Enhancement of these parameters indicate that the feedbacks between microphysics and long-wave cooling are stronger in Fog2 than in Fog1. In addition, as shown in Fig. 9g-h, vertical mixing transports fog water from the fog top to the fog base, and the strength of this transportation is stronger in Fog2 than in Fog1, because of stronger turbulent

kinetic energy (TKE) in Fog2. The effect of TKE on fog is analysed in Sect. 5.3.





## 5.3. Feedbacks between macrophysics, radiation, and turbulence

Section 5.2 analyses the microphysics-related mechanisms of self-enhanced AFIs. This subsection not only focuses on macrophysics and its feedbacks with radiation and turbulence but also discusses how the combined effects of aerosols and meteorological conditions impact

the feedbacks and enhance AFIs in Fog2, compared with those in Fog1. Briefly, fog macrophysics involves duration and distribution. The reason the duration of Fog2 is longer than that of Fog1 is related to the earlier formation of Fog2 induced by the more conducive meteorological conditions, as discussed in Sect. 5.1. The reason for the wider distribution (fog-top height and fog area) is discussed here.

### 295 5.3.1 Effects of macrophysics on radiation

The more conducive meteorological conditions and AFIs promote condensation near the fog top (Fig. 9d, f), thereby raising the fog-top height in Fog2 compared with that in Fog1 (black and purple lines in Fig. 9). Therefore, both fog-top height and FOD in Fog2 are higher than those in Fog1. Compared with that in Fog1, the higher FOD in Fog2 can enhance cooling near

the fog top and downwelling long-wave radiation, weakening the cooling at the fog base than near the fog top (Fig. 9c). Additionally, the horizontal distribution of Fog2 is wider than that in Fog1 (Fig. 5d). So, more foggy grid cells show more radiative cooling near the fog top and downwelling long-wave radiation at the fog base in Fog2.

### 5.3.2 Effects of radiation on turbulence

The above analysis reveals the mechanism of the effects of meteorology and AFIs on radiation in fog. How does radiation affect stability and turbulence (i.e., TKE)? To answer this question, we must know the dominant factors contributing to TKE, as described in the following TKE budget equation:



$$\frac{\Delta TKE}{\Delta t} = TKE_{shear} + TKE_{buoy} - TKE_{diss} + TKE_{mixing} \tag{3}$$

where $\Delta TKE/\Delta t$ is the TKE tendency with time (Fig. 10b), and the four terms on the right side of Eq. (3) are contributors to TKE, including wind shear (Fig. 10c), buoyancy (Fig. 10d), dissipation (Fig. 10e), and vertical mixing (Fig. 10f). Detailed equations of these contributions to TKE are provided in supplementary information (Eqs. S5-S8) (Nakanishi and Niino (2009)).

As shown in Fig. 10a, TKE in Fog2 is stronger than that in Fog1, particularly under
polluted conditions. As the vertical mixing term is one order smaller than the others, it is negligible (Fig. 10f). At night, only the shear term is positive and, therefore, the main contributor to TKE (Fig. 10c), consistent with the speculations of Kim and Yum (2012). However, the dominant term driving the differences in TKE between polluted and clean conditions is buoyancy (Fig. 10d). As shown in Fig. 10b, $\Delta TKE/\Delta t$ is larger under polluted
conditions than under clean conditions. Meanwhile, the shear term is smaller but the buoyancy term is larger under polluted conditions than under clean conditions; moreover, the dissipation term is similar between the two conditions. Therefore, the buoyancy term is the main factor increasing TKE under polluted conditions, corroborating the qualitative speculations by Jia et al. (2019). This is particularly true for Fog2. In addition, at daytime, $\Delta TKE/\Delta t$ is weaker under
polluted conditions, because higher FOD reduces short-wave radiation reaching the surface. These results are consistent with the stronger stability during the dissipation stage under polluted conditions, as described in Sect. 5.1.

After confirming the importance of the buoyancy term, we analyse the effect of radiation on buoyancy and then on TKE. Buoyancy contributions to TKE are determined by temperature
inversion in the PBL at the night time. As shown in Fig. 11a-b, temperature inversion is close to the surface. With the effect of AFIs, much stronger radiative cooling leads to a more rapid temperature drop at the fog top than at the fog base (Fig. 11c), thereby causing weaker





temperature inversion under polluted conditions. Therefore, stability is weaker and TKE is larger under polluted conditions, particularly in Fog2.

### 5.3.3 Effects of turbulence on macrophysics

Previous observations (Liu et al., 2010; Román-Cascón et al., 2016) and large eddy simulations (Bergot, 2013; Mazoyer et al., 2017; Nakanishi, 2000) showed that turbulence could increase the fog-top height. In this paper, we note that increasing TKE increases fog-top height (black and purple lines in Fig. 9) and fog area (Fig. 5d), which is consistent with observations of Jia et al. (2019) and Quan et al. (2021). The increased fog-top height increases TKE by promoting radiative cooling near the fog top and weakening temperature inversion. This reflects the feedbacks between macrophysics, radiation, and turbulence. Overall, due to the more conducive meteorological conditions, the feedbacks are stronger in Fog2 than in Fog1, resulting in self-enhanced AFIs.

## 6    Conclusion

To explore AFIs on the fog macro- and microphysics and their self-enhanced mechanisms, WRF-Chem 4.1.3 is used to simulate two successive radiation fog events, which occurs in the northern YRD region in China on 26 and 27 November 2018. The two fog events simulation (Fog1 and Fog2) can well reproduce the observed results.

The results show higher LWC, higher $N_f$, smaller $R_e$, higher fog-top height, longer duration, wider spatial distribution, higher LWP, and higher FOD under polluted conditions than under clean conditions. The effects of aerosols on these micro and macro-physical properties are more significant in Fog2 than in Fog1. Therefore, AFIs are self-enhanced, that is, AFIs in Fog1 enhance AFIs in Fog2. A conceptual diagram is proposed to describe the mechanism of self-enhanced AFIs (Fig. 12). Moreover, the mechanisms of AFIs are discussed based on the



synergistic effects of aerosols and meteorological conditions. In Fog1, the microphysics–radiation feedbacks and macrophysics–radiation–turbulence feedbacks delay Fog1 dissipation, generating more conducive meteorological conditions and promoting the earlier formation of Fog2. Furthermore, the microphysics–radiation feedbacks and macrophysics–radiation–turbulence feedbacks are strengthened in Fog2, enhancing AFIs in Fog2 compared with those in Fog1. Detailed mechanisms are summarised below, including meteorological conditions and the two types of feedbacks.

First, meteorological conditions before Fog2 formation are more conducive than those before Fog1 formation, which play fundamental roles in self-enhanced AFIs. This is related to the delayed dissipation of Fog1 induced by AFIs. During Fog1 dissipation (daytime), the cooling effect caused by the higher FOD contributes to the lower temperature, higher relative humidity, and stronger stability. At night, cold advection near the ground is enhanced. Meanwhile, affected by the daytime temperature, the temperature remains low, forming a cold centre. Moreover, the surface wind diverges from the cold centre to the outside, strengthening the cold advection. Ultimately, more conducive meteorological conditions induced by aerosols promote the earlier formation and longer duration of Fog2 than of Fog1.

Second, the positive feedbacks between microphysics and radiative cooling are crucial physical mechanisms for self-enhanced AFIs. In Fog2, aerosols and more conducive meteorological conditions synergistically promote fog microphysics. Lower temperature and higher relative humidity promote aerosol activation and condensation. Consequently, $N_f$, LWP, and FOD are higher, whereas $R_e$ is smaller, in Fog2 than in Fog1. These variations in microphysics lead to stronger long-wave radiative cooling and condensational growth near the top of Fog2. Therefore, the positive feedbacks between microphysics and radiation are stronger in Fog2, which further promote stronger AFIs.





Finally, the feedbacks between fog macrophysics, radiation, and turbulence affect self-enhanced AFIs. Under polluted conditions, the higher fog top strengthens the fog-top long-wave radiative cooling and then reduces the strength of temperature inversion near the surface and enhances turbulence. Stronger turbulence further increases the fog-top height and fog area. Due to more conducive meteorological conditions, the feedbacks are stronger in Fog2 than in

Fog1, contributing to self-enhanced AFIs.

   In conclusion, during the two successive radiation fog events study, AFIs are self-enhanced. Besides, there are large uncertainties in the aerosol–cloud interactions (Fan et al., 2016; Guo et al., 2018; Rosenfeld et al., 2019; Seinfeld et al., 2016; Zhu and Penner, 2020; Zhu et al., 2019). The findings in our paper shed new light on whether self-enhanced mechanisms

are at play in aerosol–cloud interactions, particularly for stratus, which is similar to fog.

   Data and code availability. The data repositories have been listed in Sect. 2. Codes can be

accessed by contacting Chunsong Lu via luchunsong110@gmail.com.

   Author contributions. NS performed the data analysis, model simulation, and paper writing. CL proposed the idea, supervised the work and revised the paper. XJ and YW both took part in revising the paper and gave suggestions. Ground-based observation data were provided by XJ

and DL. YL supervised the analysis of the turbulence kinetic energy budget. TZ supported the work that anthropogenic emissions were driven by Multiresolution Emission Inventory for China (MECI). SN provided financial support. NS prepared the paper with help from YY, BZ, SF, SY, and JL.





Competing interests. The authors in this paper declare that they have no conflict of interest with others.

Acknowledgements. We are grateful to the High Performance Computing Centre of Nanjing
University of Information Science and Technology for doing the numerical calculations in this work on its blade cluster system. This paper is also supported by the National Key Scientific and Technological Infrastructure project "Earth System Science Numerical Simulator Facility" (EarthLab).

Financial support. This research has been supported by the National Key Research and Development Program of China (2019YFA0606803)  and the National Natural Science Foundation of China (grant nos. 42027804, 41775134, 41975181,42205072).



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





**Table 1.** Summary of major parameterisation schemes.

| Scheme | Option |
|---|---|
| Microphysics | Morrison |
| Boundary layer | MYNN |
| Shortwave radiation | Goddard |
| Longwave radiation | RRTMG |
| Cumulus | Grell 3D |
| Aerosol chemistry | MOSAIC (4 bins) |
| Gas phase chemistry | CBMZ |





**Table 2.** Evaluation of PM$_{2.5}$ mass concentration. NMB, NME, MFB, and MFE stand for normalised mean bias, normalised mean error, mean fractional bias, and mean fractional error, respectively. Time '2514' (DateHour) indicates 14:00 local standard time (LST) (LST = Universal Time Coordinated + 8 h) on 25 November 2018. The other time expressions follow the same logic.

| DateHour | NMB(%) | NME(%) | MFB(%) | MFE(%) |
|---|---|---|---|---|
| 2514-2614 | 13 | 25 | 13 | 24 |
| 2614-2714 | 38 | 42 | 35 | 38 |
| Total | 25 | 30 | 24 | 28 |







**Table 3.** Quantitative estimation of AFI strength in two fog events (Fog1 and Fog2), including the responses of fog optical depth (FOD), liquid water path (LWP), and fog effective radius ($R_e$) to the changes in fog droplet number concentration ($N_f$). The ratio is the relative change between Fog1 and Fog2, calculated as (Fog2 − Fog1)/Fog1.

|  | $\Delta\ln FOD/\Delta\ln N_f$ | $\Delta\ln LWP/\Delta\ln N_f$ | $-\Delta\ln R_e/\Delta\ln N_f$ |
|---|---|---|---|
| Fog1 | 0.98 | 0.76 | 0.22 |
| Fog2 | 1.32 | 1.08 | 0.24 |
| Ratio | 34.7% | 42.1% | 9.1% |







**Table 4.** Average 2 m relative humidity ($RH_{2m}$) and boundary layer height (PBLH) above the
ground in domain 03 during 12:00–20:00 local standard time (LST) (LST = Universal Time
Coordinated + 8 h) on 25 and 26 November 2018 under clean and polluted conditions. DIF is
the difference in each property between 25 and 26 November.

|  | Clean | | | Polluted | | |
|---|---|---|---|---|---|---|
|  | Nov.25th | Nov.26th | DIF | Nov.25th | Nov.26th | DIF |
| $RH_{2m}$ (%) | 76 | 80 | 4 | 76 | 82 | 6 |
| PBLH (m) | 669 | 610 | -59 | 670 | 578 | -92 |

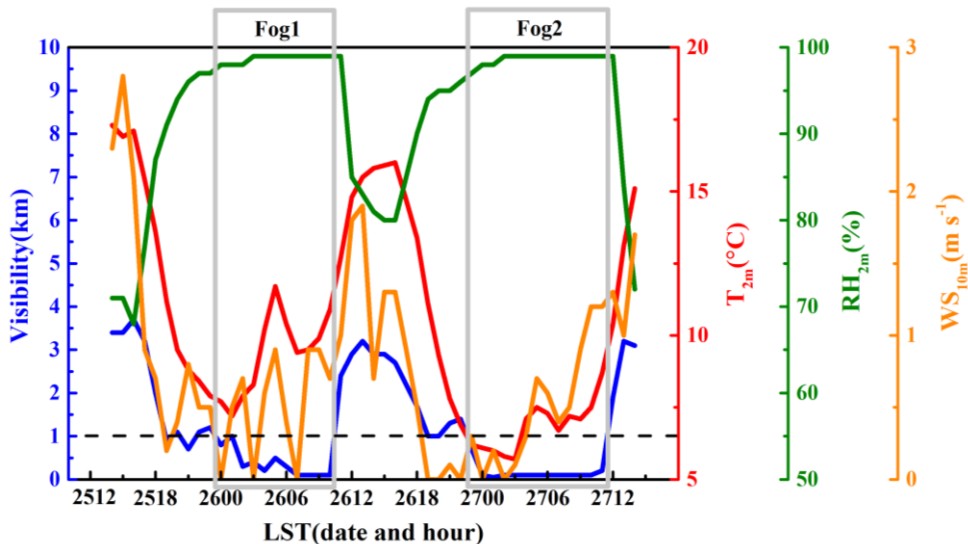


**Figure 1.** Time series of visibility, 2 m temperature ($T_{2m}$), 2 m relative humidity ($RH_{2m}$), and 10 m wind speed ($WS_{10m}$) above the ground at the Nanjing observation site (31.93°N, 118.9°E). Fog1 and Fog2 in the light grey box are the two fog events. Time '2512' indicates 12:00 local standard time (LST) (LST = Universal Time Coordinated + 8 h) on 25 November 2018. The

other time expressions follow the same logic.



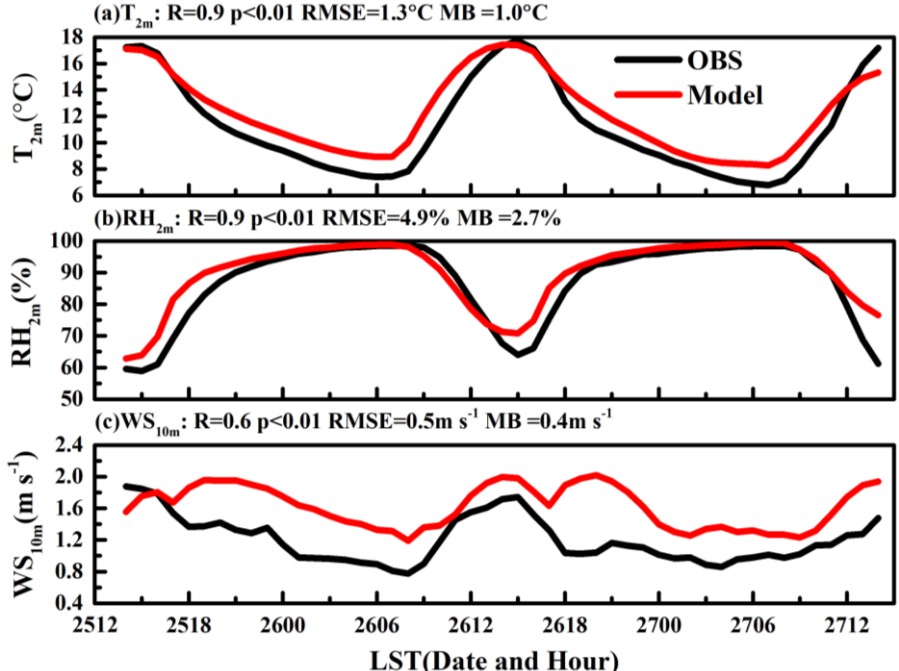

**Figure 2.** Hourly variations in observed (black lines) and simulated (red lines) meteorological properties, including (a) 2 m temperature ($T_{2m}$), (b) 2 m relative humidity ($RH_{2m}$), and (c) 10 m wind speed ($WS_{10m}$) above the ground, averaged over 104 meteorological stations in domain 03 from 14:00 local standard time (LST) (LST = Universal Time Coordinated + 8 h) on 25 November to 14:00 LST on 27 November 2018. R, p, RMSE, and MB indicate the correlation coefficient, significance level, root-mean-square error, and mean bias, respectively. Time '2512' indicates 12:00 LST on 25 November 2018. The other time expressions follow the same logic.


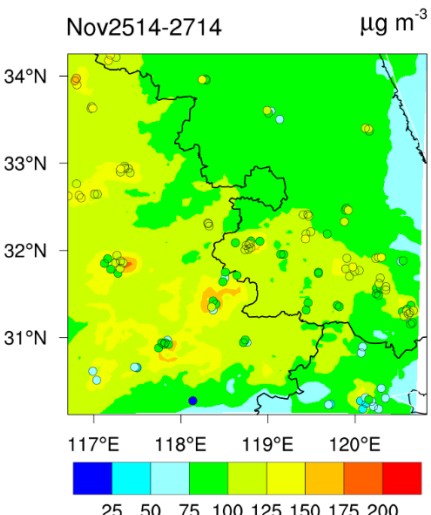

**Figure 3.** Simulated (shaded area) and observed (coloured dots) average distributions of PM$_{2.5}$ concentration (μg m$^{-3}$) from 14:00 local standard time (LST) (LST = Universal Time Coordinated + 8 h) on 25 November to 14:00 LST on 27 November 2018.




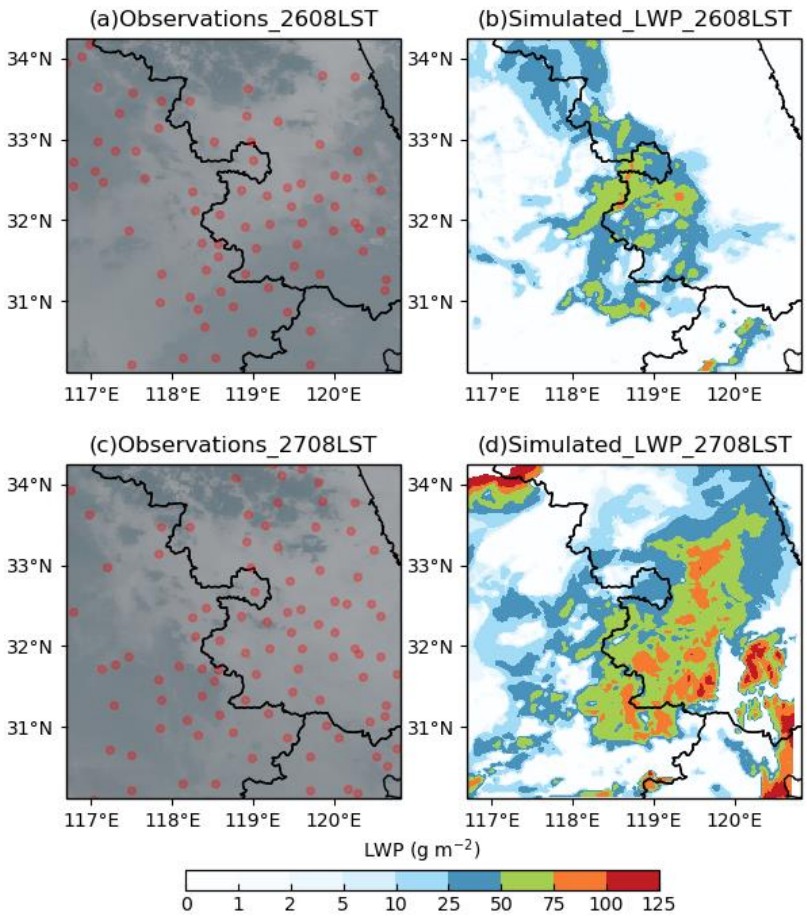

**Figure 4.** (a, c) Visible light images of Himawari-8 and ground meteorological stations with fog observations from ground sites (red dots). (b, d) Simulated liquid water path (LWP) distributions for each fog grid in domain 03. Time '2608LST' indicates 08:00 local standard time (LST) (LST = Universal Time Coordinated + 8 h) on 26 November 2018. The other time expressions follow the same logic.



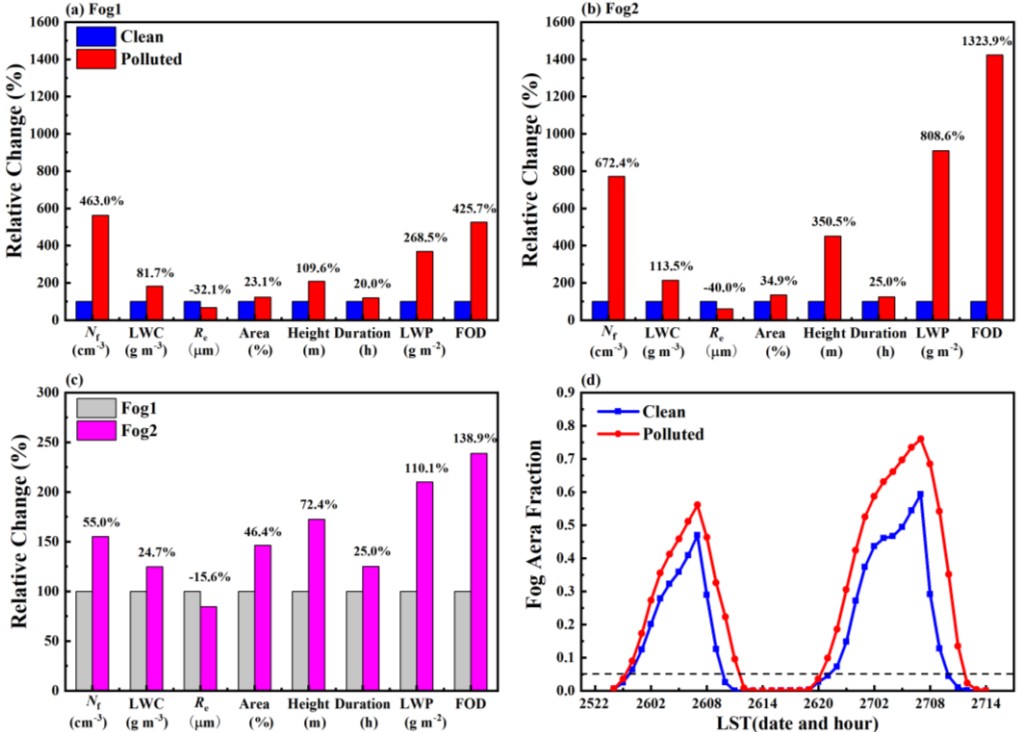

**Figure 5.** Relative changes in macro- and microphysical properties in (a) Fog1 and (b) Fog2 under polluted conditions *versus* clean conditions. (c) Relative changes in macro- and microphysical properties in Fog2 *versus* Fog1 under polluted conditions. (d) Temporal evolution of fog area fraction under clean and polluted conditions. $N_f$, LWC, $R_e$, Area, Height, Duration, LWP, and FOD indicate fog number concentration, liquid water content, effective radius, fog area fraction, fog top height, liquid water path, and fog optical depth, respectively. The properties are set to be 100% under clean conditions in Fig. 5a–b; similarly, the properties during Fog1 in Fig. 5c are set to be 100%. Numbers above the bars represent the corresponding relative changes calculated as (Polluted − Clean)/Clean in Fig. 5a-b and as (Fog2 − Fog1)/Fog1 in Fig. 5c. Time '2522' in Fig. 5d indicates 22:00 local standard time (LST) (LST = Universal Time Coordinated + 8 h) on 25 November 2018. The other time expressions follow the same logic.



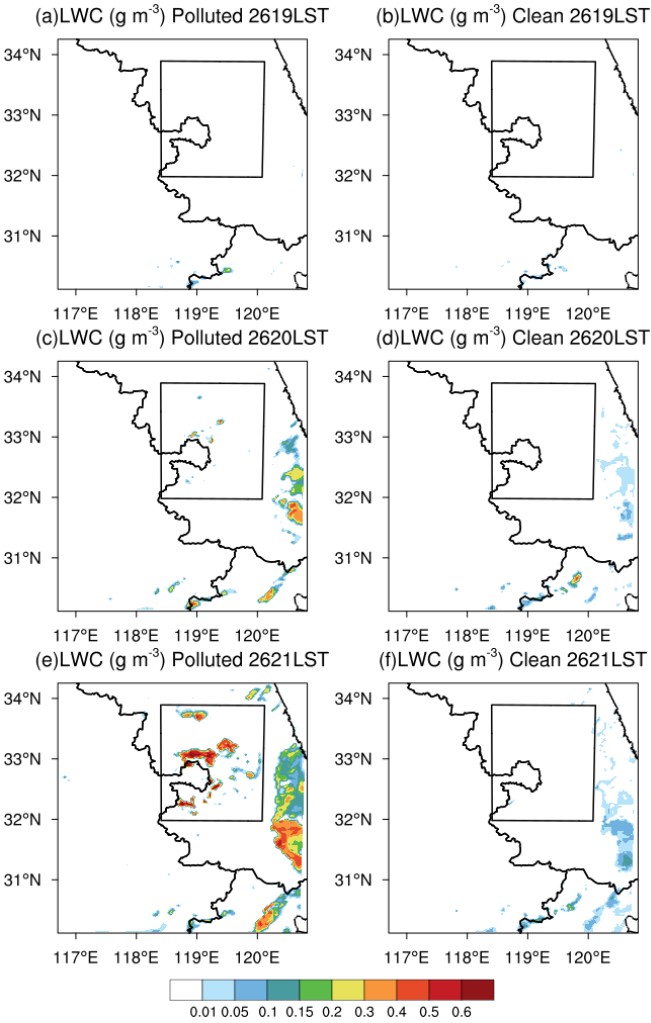

**Figure 6.** Liquid water content (LWC) distribution at the bottom layer from 19:00-21:00 local standard time (LST) (LST = Universal Time Coordinated + 8 h) on 26 November 2018 under (a, c, e) polluted and (b, d, f) clean conditions. The black box is the area in which Fog2 formed earlier under polluted condition. Time '2619LST' indicates 19:00 LST on 26 November 2018. The other time expressions follow the same logic.



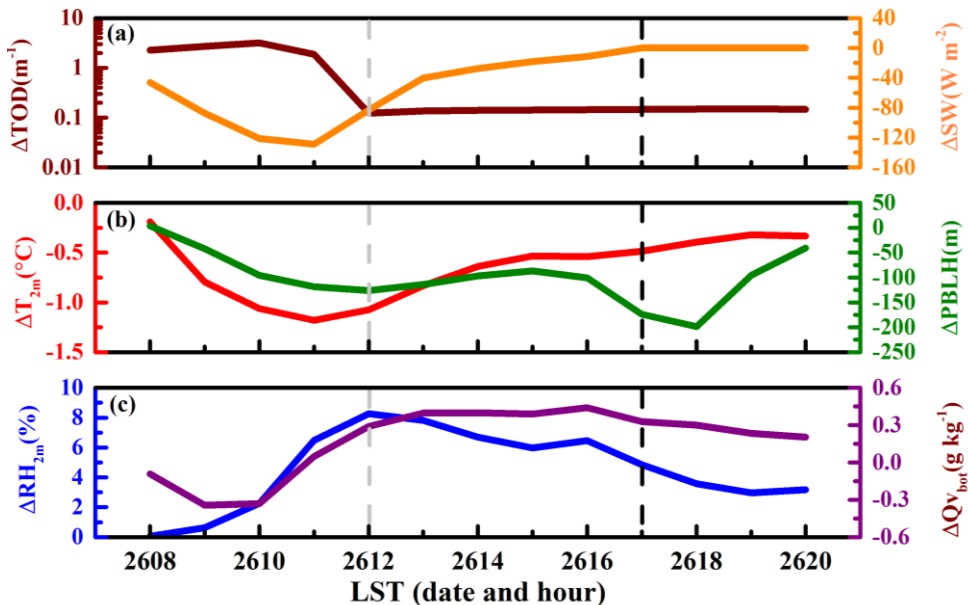

**Figure 7.** Differences in properties between polluted and clean conditions in the black box in
Fig. 6, including (a) total optical depth (TOD), surface downwelling shortwave radiation (SW),
(b) 2 m temperature ($T_{2m}$), planetary boundary layer height (PBLH), (c) 2 m relative humidity
($RH_{2m}$), and water vapour mixing ratio at the bottom of the model ($Qv_{bot}$), where TOD = FOD
(fog optical depth) + AOD (aerosol optical path). Grey dashed line is the time of complete
evaporation of Fog1 under polluted conditions. Black dashed line is the time of sunset. Time
'2608' indicates 08:00 local standard time (LST) (LST = Universal Time Coordinated + 8 h)
on 26 November 2018. The other time expressions follow the same logic.

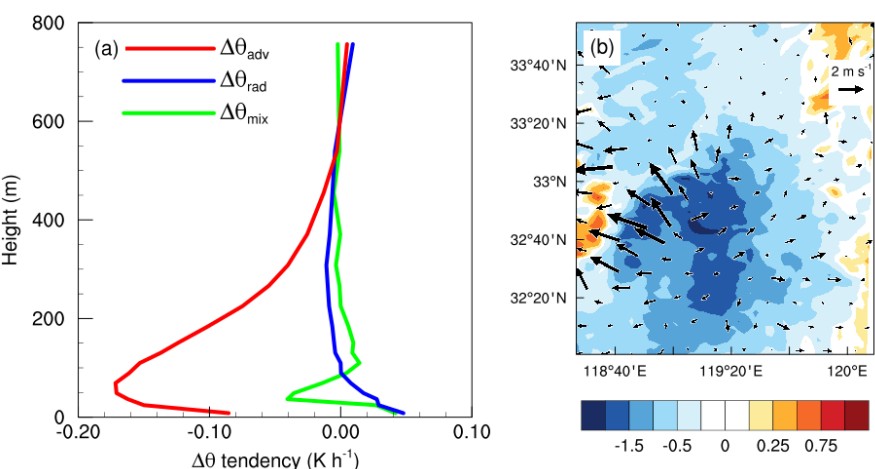

**Figure 8.** (a) Differences (Polluted − Clean) in terms contributing to the potential temperature tendency, including radiation ($\theta_{rad}$), vertical mixing ($\theta_{mix}$), and advection ($\theta_{adv}$) in the black box in Fig. 6 before fog formation (17:00–19:00 local standard time [LST = Universal Time Coordinated + 8 h]). (b) The shaded area is the mean temperature difference (Polluted − Clean), and vectors are the mean wind vector difference (Polluted − Clean) at the bottom of the model.

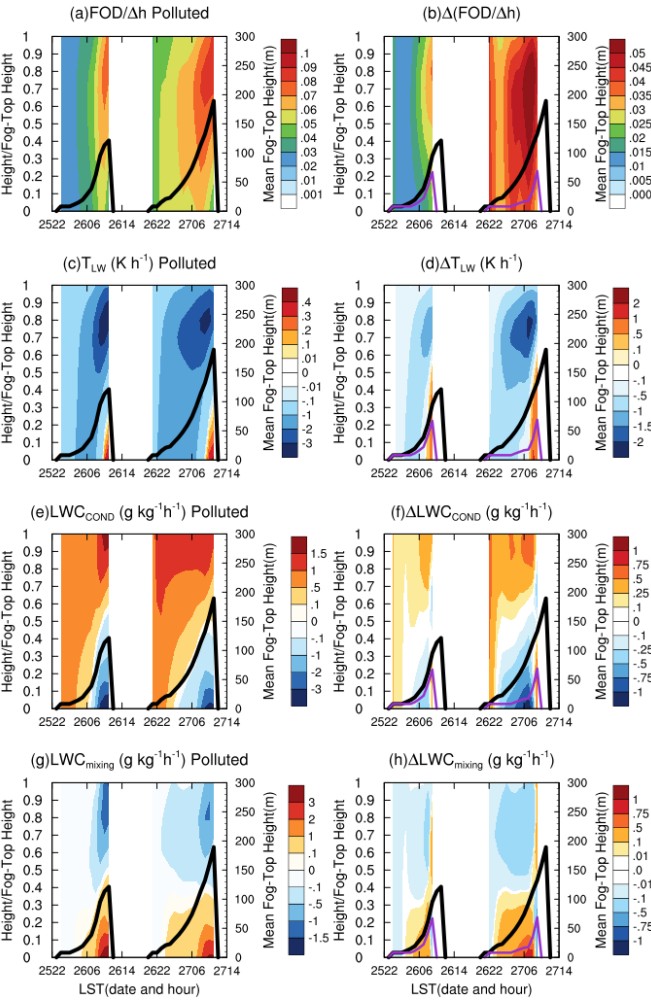

**Figure 9.** Time-height profiles of (a-b) fog optical depth per unit height (FOD/Δh), (c-d) radiative cooling rate ($T_{LW}$), (e-f) condensation growth rate ($LWC_{COND}$), and (g-h) liquid water content tendency due to vertical mixing ($LWC_{mixing}$). Heights on the left axes are normalised by the fog-top heights and the left axes are mean fog-top heights. The left column is polluted conditions and the right one is the difference (Polluted − Clean). Black and purple lines are the mean fog top heights under polluted and clean conditions, respectively. Time '2522' indicates 22:00 local standard time (LST) (LST = Universal Time Coordinated + 8 h) on 25 November 2018. The other time expressions follow the same logic.




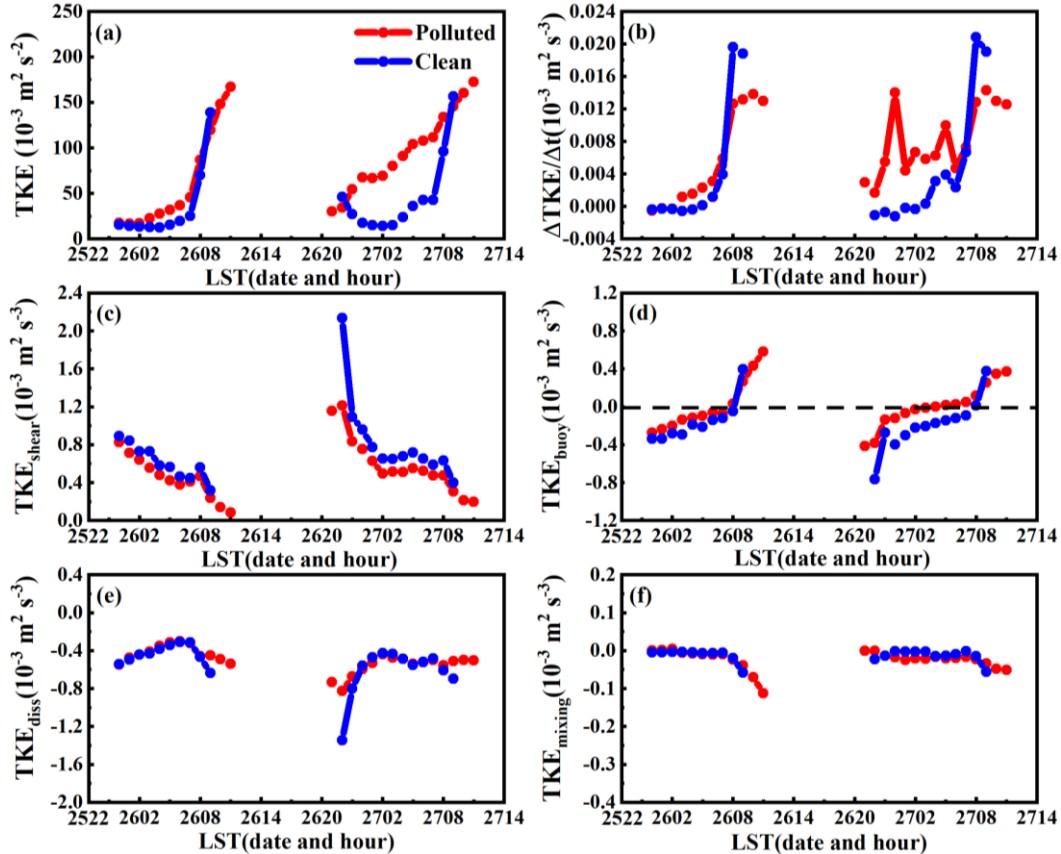

**Figure 10.** (a) Temporal evolution of turbulent kinetic energy (TKE), (b) TKE tendency, (c) wind shear term (TKE$_{shear}$), (d) buoyancy term (TKE$_{buoy}$), (e) dissipation term (TKE$_{diss}$), and (f) vertical mixing terms (TKE$_{mixing}$) under polluted and clean conditions. Dashed line is the zero

line for TKE$_{buoy}$. Time '2522' indicates 22:00 local standard time (LST) (LST = Universal Time Coordinated + 8 h) on 25 November 2018. The other time expressions follow the same logic.





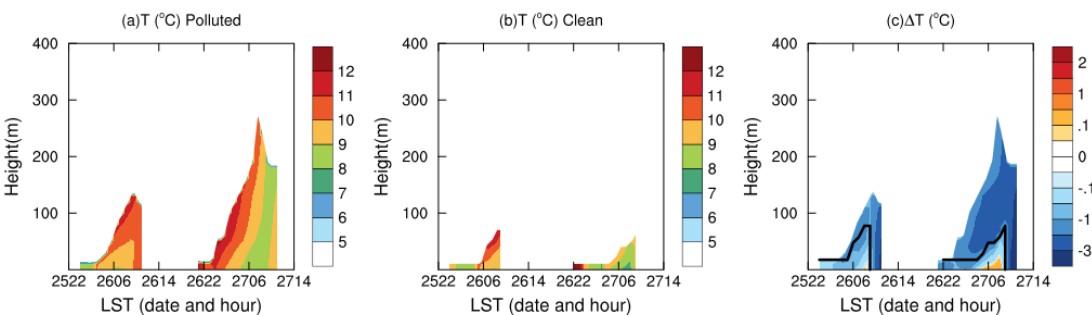

**Figure 11.** Time-height profiles of in-fog temperature (T) under (a) polluted and (b) clean conditions. (c) Difference between polluted and clean conditions. Black line on the right side is the maximal fog-top height under clean conditions. Time '2522' indicates 22:00 local standard time (LST) (LST = Universal Time Coordinated + 8 h) on 25 November 2018. The other time expressions follow the same logic.


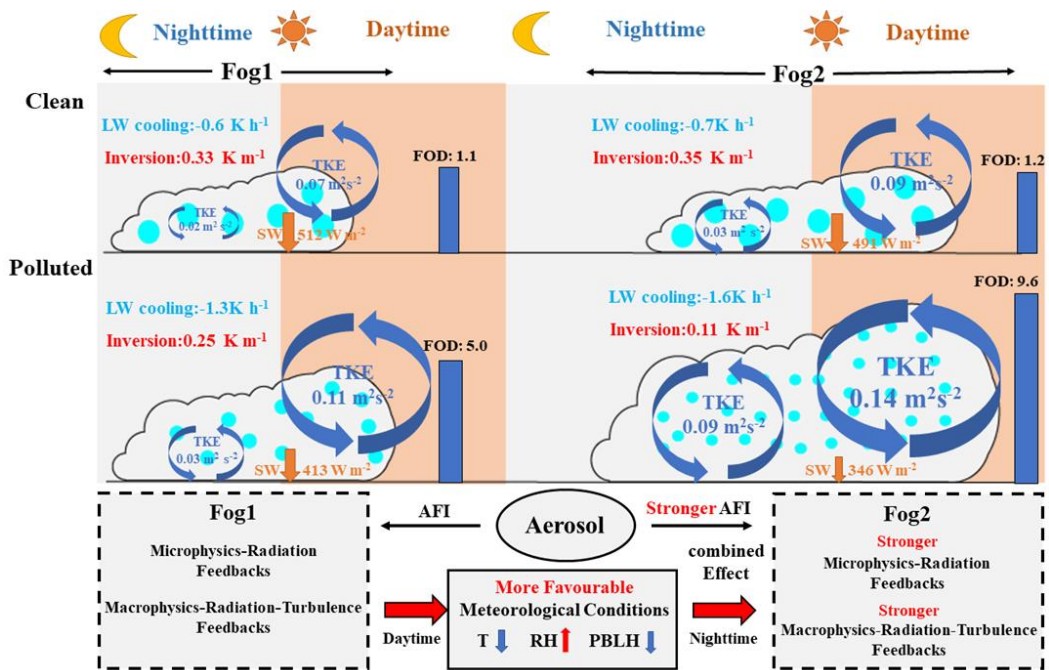


**Figure 12.** Conceptual image of self-enhanced aerosol–fog interactions (AFIs). FOD, SW, LW, TKE, T, RH, and PBLH stand for fog optical depth, short-wave radiation, long-wave radiation, turbulent kinetic energy, temperature, relative humidity, and planetary boundary layer height, respectively. LW and inversion are at night time, and FOD is at daytime.
