# Peer review of "Radiation fog properties in two consecutive events under polluted and clean"

_Atmospheric Chemistry and Physics, 2022_

## Author Comment (AC1)

**Response to Referee #1**

Dear Referee,

We appreciate your positive and constructive comments. We have read these comments carefully and made revisions accordingly. The responses to the comments are listed below.

Sincerely,

Naifu Shao, Chunsong Lu*, and all co-authors.

**Main comments:**

This manuscript presents a modelling study of two successive fog events in the Yantze River Delta region of China. It aims to show how the fog properties in the second event are influenced by the first. I find this a truly interesting topic and exciting approach. However, I struggled with the manuscript for the following reasons:

**Response**: Thank you for your valuable comments.

1.    The central message is that fog properties are influenced by aerosol as well as other boundary-layer conditions. The latter may be modified by a preceding fog event, resulting in fog property differences between both events. (a) This simple -- and very interesting – finding is hidden behind the phrase "self-enhanced AFIs", and thus took me more time to understand than would have been necessary. I would suggest to focus on the changes to the fog rather than "AFIs", and to speak about "aerosol loading" or "polluted conditions" to clarify the meteorological context. (b) Also, "AFIs", which is modelled on the common abbreviation "ACI" for aerosol-cloud interactions should probably lose the "s" to make it consistent with ACI. (c) A change of the title could also

be considered to more close reflect the paper's focus, e.g. "Radiation fog properties in two consecutive events under clean and polluted conditions..." or similar.

**Response**: Thank you for your suggestion.

(a) We agree with the referee and have deleted the phrase "Self-enhanced AFI" in many places of the abstract and main text (e.g., Page 1 Line 24; Page 5 Lines 111-113). Instead, we focus more on the changes to the fog, as suggested by the referee. We also speak about "aerosol loading" or "polluted conditions" to clarify the meteorological context. For example:

- "Our simulations indicate that conducive PBL conditions are affected by AFI with high aerosol loading in Fog1, and then PBL promotes AFI in Fog2, resulting in higher liquid water content, higher droplet number concentration, smaller droplet size, larger fog optical depth, wider fog distribution, and longer fog lifetime in Fog2 than in Fog1" (Page 1, Lines 20-24).

- "The two fog scenarios provide an excellent opportunity to analyse AFI under polluted conditions as a chain, i.e., how high aerosol loading affects properties in the first fog scenario, how the properties in the first polluted fog scenario affect radiation and the PBL structure, and then how radiation and the PBL affect properties and AFI in the second fog scenario under polluted conditions" (Page 5, Lines 101-105).

- "Here, we study how radiation fog properties are affected by high aerosol loading and PBL meteorological conditions in two successive events in the YRD region" (Page 5, Lines 115-116).

- "Furthermore, compared with the difference of aerosol-induced changes in $RH_{2m}$ and PBLH before fog formation, $RH_{2m}$ increases by 6% and PBLH decreases by 92 m under polluted conditions, which is larger than those ($RH_{2m}$: 4% and PBLH: -59 m) under clean conditions" (Pages 11-12, Lines 266-269).

(b) All the "AFIs" in this manuscript are revised to "AFI".

(c) According to your suggestion, we have revised the title to be "Radiation fog properties in two consecutive events under polluted and clean conditions in the Yangtze River Delta, China: A simulation study".

2. (a)The state of the art chapter does not seem complete. The central motivation, i.e. limited knowledge about AFI, is only briefly stated, and not explained (line 81). (b) The fundamental premise that an event may be influenced by a previous event does not follow from the literature review presented at all. (c) The focus, concepts and terminology of the first research question are neither derived from the literature, nor are they explained. (1) What is a "stronger" fog scenario? (2) What does "stronger AFIs" mean? (3) What would you expect? Why? And why does it matter?

**Response**: Thank you for your suggestion.

(a) Regarding limited knowledge about aerosol–fog interaction (AFI), we meant that it is not clear how AFI and planetary boundary layer (PBL) interacts with each other and the evolution of AFI in successive fog scenarios remains unknown. To make the description clearer, we pointed out the questions directly (Page 4, Lines 95-99): "What are the physical mechanisms behind the property changes during the two successive fog events? Furthermore, which fog scenario has fog macro- and microphysical properties more sensitive to aerosol, i.e., experiencing stronger AFIs? Are the mechanisms related to the interaction between AFI and PBL?"

(b) We have added the sentences to show that an event may be influenced by a previous event (Pages 4, Lines 92-95): "Previous studies typically focused on an individual fog event or analysed multiple fog events statistically, however, there were still several studies mentioning that LWC, $N_f$ and liquid water path (LWP) in the latter fog scenario were larger than those in the preceding one (Quan et al., 2011; Wærsted et al., 2017)."

(c) The focus, concepts and terminology are explained as follows.

(1) "Stronger fog scenario" means a fog scenario has larger macro- and microphysical

properties, such as fog droplet number concentration and liquid water content. To be more specific, we have improved the description (Pages 4, Lines 93-94): "liquid water content, droplet number concentration and liquid water path in the latter fog scenario were larger than those in the preceding one".

(2) "Stronger AFIs" means the more remarkable fog property response to changes in aerosol loading. For example, if aerosol-induced changes in fog optical depth is larger, AFI is stronger. We have added the above explanation (Page 4, Lines 96-98): "which fog scenario has fog macro- and microphysical properties that are more sensitive to aerosol, i.e., experiencing stronger AFI?"

(3) The reason to analyze the evolution of AFI in two fog scenarios is that stronger AFI can affect fog development, for example, increasing droplet number concentration more significantly. Furthermore, we would like to examine the mechanisms responsible for the evolution of AFI and study how the interaction between AFI and PBL make fog properties change in the two successive fog events. We have revised the manuscript accordingly:

- However, it is not clear how AFI in the first fog (Fog1) affects PBL and then AFI in the second fog (Fog2), which is important to understand the interaction between AFI and PBL as well as their effects on fog properties (Page 1, Lines 16-18).

- Our simulations indicate that conducive PBL conditions are affected by AFI with high aerosol loading in Fog1, and then PBL promotes AFI in Fog2, resulting in higher liquid water content, higher droplet number concentration, smaller droplet size, larger fog optical depth, wider fog distribution, and longer fog lifetime in Fog2 than in Fog1 (Page 1, Lines 20-24).

3. In some places, aspects concerning methodology and interpretation remain unclear. (a)How precisely is the validation performed? (b)To what extent and under what conditions can the findings of this study be generalized? (c)Instead of using "AFIs", in

many places it would be more helpful to explicitly address the parameter of relevance, e.g. LWP, aerosol loading, droplet radius...

**Response**: Thank you for your suggestion.

(a) We add Table 3 to explain the elements a–d in the Heidke skill (HSS) score. In our study, the HSS score are 0.34 and 0.36 in Fog1 and Fog2, respectively, which are close to previous reports (Mecikalski et al., 2008; Xu et al., 2020; Yamane et al., 2010). We have added the above description (Page 9, Lines 198-200).

**Table 3.** The elements a-d in the Heidke Skill Score calculation

|  | Fog observed | No fog observed |
|---|---|---|
| Fog simulated | a | b |
| No fog simulated | c | d |

(b) "Our findings can be generalized due to the following reasons. First, the simulation design is reasonable. Similar to many previous studies, polluted and clean conditions are simulated through varying emission intensity. Second, the conclusions are robust, because they are derived from physical analyses. The interactions between aerosol loading, fog macro- and microphysical properties, and boundary layer meteorological conditions are understood physically. Third, the fog events are typical and have large coverage. Therefore, the findings in this study can be generalized, at least in polluted fog events during winter." The above discussions are added (Page 19, Lines 444-450).

(c) Thank you for your suggestion. AFI is replaced by parameters of relevance.

- "Larger TOD, particularly larger FOD, leads to lower SW, $T_{2m}$, and PBLH" (Page 12, Lines 277-278).

- "Larger FOD and delaying dissipation result in lower temperature, higher relative humidity, and higher stability by affecting solar radiation during the daytime" (Page 13, Lines 293-294).

- "The cold centre is related to lower temperature under polluted conditions due to

larger FOD and longer duration in Fog1" (Page 13, Lines 298-300).

4.   While the paper is both legible and intelligible, it would profit from a linguistic revision.

Response: Thank you for your comment. Hope this manuscript has been improved after a linguistic revision.

**DETAILS**

(1) 15 - "pivotal" is unclear here

**Reply**: The word "pivotal" is replaced by "critical" (Page 1, Line 16).

(2) 15 - what is "the fog cycle"?

**Reply**: We mean the fog life cycle. We have revised the sentence: "Aerosol–fog interaction (AFI) and planetary boundary layer (PBL) conditions play critical roles in the fog life cycle" (Page 1, Lines 15-16).

(3)  16: Why should they focus on these differences? What is special about successive events?

(4) 17: What knowledge gap exactly?

**Reply**: We would like to reply to the two comments together, because they are closely related to each other. The difference between two successive events is important to understand the interaction between Aerosol–fog interaction (AFI) and planetary boundary layer (PBL) as well as their effects on fog properties. That is why we are interested in the difference between two fog events. However, it is not clear how AFI in the first fog affects PBL and then AFI in the second fog. This is the knowledge gap.

We have revised the abstract accordingly (Page 1, Lines 16-18).

(5) 19: "AFIs ... promote..." -- Do you mean high/low aerosol loadings? Or the interaction (mechanisms) specifically?

**Reply**: We mean the interaction (mechanisms) specifically. We have revised the sentence (Page 1, Lines 20-24): "Our simulations indicate that conducive PBL conditions are affected by AFI with high aerosol loading in Fog1, and then PBL promotes AFI in Fog2, resulting in higher liquid water content, higher droplet number concentration, smaller droplet size, larger fog optical depth, wider fog distribution, and longer fog lifetime in Fog2 than in Fog1."

(6) 22: "is defined as" -- you mean that you define it as, or is this taken from elsewhere?

**Reply**: We mean that we define it as. This phrase is deleted because self-enhanced AFI is deleted, according to the referee's other comments.

(7) 38: fog does not lead "to environmental pollution" - please clarify this statement

**Reply**: We agree with the referee and have deleted this phrase (Page 2, Lines 40-41): "This leads to low visibility, affecting human health, transportation, and power system (Niu et al., 2010)".

(8) 40: You state that the "physical processes of fog remain unclear". What exactly do you refer to? Can you provide a reference, please? I would think that the processes are pretty well understood.

**Reply**: We have reorganized the sentences to describe the unclear physical processes of fog and have added references (Page 2, Lines 41-46): "There exist uncertainties in fog

forecasting (Zhou and Du, 2010; Zhou et al., 2011). An important reason is that the physical processes of fog remain unclear, because many processes (aerosol activation, condensation, radiation as well as turbulence) not only occur simultaneously but also interact with each other nonlinearly (Haeffelin et al., 2010), which affects fog properties (Mazoyer et al., 2022) and impedes the related parameterisation (Poku et al., 2021)".

(9) 47: First sentence is a repetition of statement in line 36.

**Reply**: We have revised the sentence (Page 3, Lines 52-53): "Since fog is a special type of cloud (Guo et al., 2021; Kim and Yum, 2010, 2013; Wang et al., 2023), AFI is expected to share similarities with aerosol–cloud interaction".

(10) 52: What do you mean by "fog number concentration"? droplet number concentration in fog?

**Reply**: Yes, we mean fog droplet number concentration. The phrase is revised accordingly (Page 3, Line 60).

(11) 53: Can these numbers be generalized? How would they be expected to change given different environmental conditions? Is this continental radiation fog, sea fog, advection fog over land, ...?

**Reply**: The referee's concern is reasonable. Here we take the two fog field campaigns as examples representing polluted and clean conditions, respectively. Although the field campaign in the North China Plain cannot fully stand for all polluted conditions and the field campaign in Xishuangbanna, China, cannot fully stand for all clean conditions, the comparison between the two examples does show the difference of fog properties between polluted and clean conditions, i.e., fog droplet number concentration is higher and effective radius is smaller in the polluted conditions than in the clean one. Examples

above are both continental radiation fog.

We have revised the sentences as follow (Page 3, Lines 57-63): "Different continental fog observation projects showed that fog microphysical properties were significantly affected by aerosol loading (Mazoyer et al., 2019; Niu et al., 2011; Quan et al., 2011; Wang et al., 2021). In those polluted fog observations, for instance, Quan et al. (2011) found that the fog droplet number concentration ($N_f$) was higher than 1,000 cm$^{-3}$ and effective radius ($R_e$) was approximately 7 μm in the North China Plain. In those clean fog observations, for example, Wang et al. (2021) showed that $N_f$ was smaller than 100 cm$^{-3}$ and $R_e$ was approximately 9 μm in the tropical rainforest in Xishuangbanna, China".

(12) 70: That radiative cooling "is an important factor for temperature inversion, providing stable conditions for fog formation" is not a finding of the cited studies in the 2010s, but can be derived from very basic textbook knowledge.

**Reply**: We agree with the referee and have revised the sentence (Page 4, Lines 77-79): "Early studies showed that radiative cooling was an important factor for temperature inversion, providing stable conditions for fog formation (Fitzjarrald and Lala, 1989; Holets and Swanson, 1981; Roach et al., 1976)".

(13) 81: In what respect is this knowledge limited? What is lacking?

(14) 83: Why do you think successive fog events are worth considering?

**Reply**: The two comments are replied together. The understanding of AFI remains limited because the mechanism behind the interaction between aerosol, fog and PBL is not fully studied, especially in two successive fog events. We have reorganized the sentences to make a clearer description (Page 4, Lines 92-99): "Previous studies typically focused on an individual fog event or analysed multiple fog events statistically, however, there were still several studies mentioning that LWC, $N_f$, and liquid water path

(LWP) in the latter fog scenario were larger than those in the preceding one (Quan et al., 2011; Wærsted et al., 2017). What are the physical mechanisms behind the property changes during the two successive fog events? Furthermore, which fog scenario has fog macro- and microphysical properties more sensitive to aerosol, i.e., experiencing stronger AFI? Are the mechanisms related to the interaction between AFI and PBL?"

(15) 84/5: Why?

**Reply**: The reason is that fog is a special cloud near ground. We have revised the sentence (Page 5, Lines 105-108): "Additionally, because fog is a special cloud near ground, the evolution of AFI is also helpful to study the evolution of aerosol–cloud interaction, which is critical to climate prediction (Boutle et al., 2018; Vautard et al., 2009)".

(16) 89: How do you define "stronger AFIs", what do you mean by this and why does it matter?

**Reply**: "Stronger AFIs" are defined as the more remarkable fog property response to changes in aerosol concentration. For example, if aerosol-induced change in fog optical depth is larger, AFI is stronger. The reason to compare AFI strength in two fog scenarios is that stronger AFI can promote fog development, for example, increasing droplet number concentration more significantly. Furthermore, we would like to examine the mechanisms responsible for the evolution of AFI and study how the interaction between AFI and PBL make fog properties change in the two successive fog events. We have revised the manuscript accordingly (Page 4, Lines 96-98, Page 5, Lines 101-105).

(17) 101: What aerosol species?

(18) 101: What is "massive"? Please be more specific.

**Reply**: We would like to reply two comments together. It is $PM_{2.5}$ and the $PM_{2.5}$ mass concentration is over 100 µg m$^{-3}$.

We have revised the sentence (Pages 5, Lines 116-118): "The $PM_{2.5}$ mass concentration was over 100 µg m$^{-3}$ before fog events in the YRD due to anthropogenic emissions (Zhu et al., 2019)".

(19) 140: What does this experiment consist of, and what sensitivities are tested for?

**Reply**: The control run is tested for the polluted conditions with emission intensity directly from the MEIC database. The sensitive experiment is tested for the clean conditions with the emission intensity multiplied by 0.05. The design of the control run and sensitivity test is the same as those in Jia et al. (2019) and Yan et al. (2020). We have revised the sentences (Pages 7, Lines 156-160).

(20) 160: How do you compare observations and model? How do you define "consistent" in this regard?

**Reply**: We use ground-based fog observations and cloud optical depth from Himawari-8 to evaluate simulations (Figure 4). "consistent" is defined as similarity of the simulated fog spatial distributions and magnitude of optical depth between simulations and observations. We add "generally" before "consistent" and revised the sentences to be more objective (Page 8, Lines 184-187): "Qualitatively, the simulated fog spatial distributions and magnitude are generally consistent with satellite and ground-based observations. Similarly, Lee et al. (2016) evaluated fog distribution simulation against cloud optical depth from satellite; they also concluded that the distributions of simulation and observation were generally comparable with each other".

Besides the qualitative evaluation, we also use HSS to quantitatively evaluate the simulations. Please see the response to the next comment.

[Figure]

Figure 4. (a, c) Distributions of ground-based fog observations (the circular points) and cloud optical depth from Himawari-8 products at 08:00 LST on 26-27 November 2018. (b, d) Simulated fog optical depth (FOD) distributions in the domain 03 at the corresponding time of observations. Time '2608LST' indicates 08:00 local standard time (LST) (LST = Universal Time Coordinated + 8 h) on 26 November 2018. The other time expressions follow the same logic.

(21) 161: Based on which parameters is HSS calculated?

**Reply**: Table 3 is added to more clearly show how HSS is calculated. The description of HSS score is also revised (Pages 8-9, Lines 191-197): "Elements $a$–$d$ are the numbers of "hits", "false alarms", "misses", and "correct negatives", respectively, which are determined by observations and simulations as shown in Table 3. To identify observed fog at a station, two criteria are used: visibility less than 1 km and relative humidity larger than 90% (Yan et al., 2020). Simulated foggy grids are recognized based on three criteria (Jia et al., 2019; Zhao et al., 2013): fog water mixing ratio over 0.01 g kg$^{-1}$, $N_f$

greater than 1 cm$^{-3}$, and the fog base touching the ground. The elements $a$–$d$ are calculated based on fog occurrence at the observation stations and the closest model grids".

Table 3. The elements a-d in the Heidke Skill Score calculation

|  | Fog observed | No fog observed |
| --- | --- | --- |
| Fog simulated | a | b |
| No fog simulated | c | d |

(22) 172: Why this threshold?

Reply: "We also test other thresholds, 1%, 2.5%, 7.5%, and 10% (Fig. S3). The results are similar to those based on the threshold of 5%." We have added the above description (Page 9, Lines 206-208). Figure 3 is added in the supplement.

[Figure]

Figure S3. Aerosol effect on relative changes in macro- and microphysical properties during the first fog (Fog1) and the second fog (Fog2). Figure S3 a-d are the results with fog fraction area thresholds 1%, 2.5%, 7.5%, and 10% respectively. $N_f$, LWC, $R_e$, Area, Height, Duration, LWP, and FOD indicate fog number droplet concentration, liquid water content, effective radius, fog area fraction, fog top height, liquid water path, and fog optical depth, respectively. The ratio of changes is calculated as Polluted/Clean.

(23) Figure 5: I find it slightly confusing that the reference case is shown with 100% bars in all cases. I suggest leaving this out and only showing the polluted (a,b) or fog2 (c) situations.

**Reply**: We agree with the referee and have deleted the 100% bars. As suggested, we only show figure (a, b) and deleted figure c.

[Figure]

**Figure 5.** (a) Aerosol-induced changes in macro- and microphysical properties during the first fog (Fog1) and the second fog (Fog2) under polluted and clean conditions. (b) Temporal evolution of fog area fraction under clean and polluted conditions. $N_f$, LWC, $R_e$, Area, Height, Duration, LWP, and FOD indicate fog droplet number concentration, liquid water content, effective radius, fog area fraction, fog-top height, liquid water path, and fog optical depth, respectively. The ratios of changes are calculated by Polluted/Clean in Fig. 5a which reveal the aerosol-induced changes. The numbers above the bars in Fig. 5a represent the difference in those ratios of changes between Fog1 and Fog2 (calculated by Fog2-Fog1). Time '2522' in Fig. 5b indicates 22:00 local standard time (LST) (LST = Universal Time Coordinated + 8 h) on 25 November 2018. The other time expressions follow the same logic.

(24) 185ff: Here, and in several other places, you assume that AFI lead to changes in fog2. In section 5 you state that fog2 is different because boundary-layer conditions are different after a previous fog event, and not specifically because of the aerosol. Please make sure your reasoning is consistent.

**Reply**: Sorry for the confusion. Fog2 formation is related with PBL conditions which can be affected by AFI. The reasoning is that AFI postpones the dissipation of Fog1 due to feedbacks in fog and generates more conducive PBL meteorological conditions before Fog2 than before Fog1; these more conducive conditions promote the earlier formation of Fog2. We have revised the whole manuscript (e.g., Page 11, Lines 246-247; Page 2, Lines 33-36).

(25) 241: higher stability

**Reply**: Stronger stability has been replaced by higher stability, according to your advice.

We have revised the sentence (Page 13, Line 302): "Therefore, lower temperature, higher relative humidity, and higher stability result from AFI in Fog1, contributing to the earlier formation of Fog2".

**References**

[revised manuscript text omitted]

---

## Author Comment (AC2)

**Response to Referee #2**

This paper describes a case study of two fog events on two consecutive days in Nanjing, simulated by WRF-chem, and proposes that aerosol-fog interactions in the first fog promote aerosol-fog interactions in the second. Most of the hypothesis is reasonable: the first fog influences boundary layer turbulence for the second fog, and that influence is affected by aerosols. However, I am not yet convinced whether the hypothesis that *aerosol-fog interaction* in the second fog is affected by aerosol-fog interaction in the first is adequately demonstrated by the simulations in the paper.

Despite this, the paper describes a useful and interesting study of aerosol-fog interactions, which in itself is well worth publishing in ACP. It is also well structured and well written, in general. I recommend that the authors either perform additional simulations to test their hypothesis, they weaken their definition of self-enhancement, or they change the message of the paper to simply highlight aerosol-fog interactions in Nanjing. Either way, in my assessment the article needs major revisions, but assuming the major comments can be addressed, it would be suitable for ACP.

Dear Referee,

Thank you for your positive and constructive comments. We have addressed your comments and the corresponding replies are listed below. Briefly, we have performed additional simulations, weakened the definition of self-enhancement and changed the message of the paper to simply highlight aerosol-fog interactions in Nanjing, according to your suggestions.

With regards,

Naifu Shao, Chunsong Lu*, and co-authors.

**Major comments**

1. The authors' summary of their evidence for their hypothesis of 'self-enhanced aerosol-fog interactions' is that by increasing droplet concentrations and by postponing the dissipation of the first fog and promoting the earlier formation of the second, aerosols increase the fog thickness and prolong its lifetime.

**Reply**: Yes, we agree with you. "AFI postpones the dissipation of Fog1 due to these two feedbacks and generates more conducive PBL meteorological conditions before Fog2 than before Fog1. These more conducive conditions promote the earlier formation of Fog2, further enhancing the two feedbacks and strengthening the AFI" (Page 2, Line 33-36).

2. Figure 7 shows the meteorological differences that arise during the first fog between clean and polluted conditions persist into the second fog. This figure is key. But would these meteorological differences still persist if the second fog, and the period between the fogs, were not also polluted? Can the authors demonstrate that direct aerosol-meteorology interactions during the clear-sky period between the two fogs do not lead to the meteorological differences in Figure 7 and the early onset of Fog 2?

**Response**: Thank you for your suggestion. We think it is possible to reply to this comment by examining the meteorological conditions before Fog 1, instead of examining the conditions before Fog 2. The reason is that there is no fog before Fog 1; all the differences of meteorological conditions before Fog 1 is caused by aerosol-meteorology interaction, which was the question the reviewer asked. "As shown in Table 5, the relative humidity at 2 m ($RH_{2m}$) above ground and planetary boundary layer height (PBLH) before Fog1 on 25 November under clean conditions are 76 % and 669 m, respectively, quite similar to those under polluted conditions (76 % and 670 m, respectively). Therefore, it is not likely that aerosol-meteorology interaction can lead to the meteorological differences in Figure 7. Besides, a previous study (Yan et al., 2021) also noted that aerosol–fog interaction was more remarkable than aerosol–

radiation interaction." The above discussions are added in the revised manuscript (Page 12, Lines 285-290).

3. Assuming the authors can demonstrate this, their theory is aerosol-fog interaction in Fog 1 changes meteorology which enhances aerosol-fog interaction in Fog 2. They show the first part of this in Figure 7: aerosol-fog interactions affect meteorology. It's reasonable that this influences the formation time of Fog 2 in the simulations. But does it also influence aerosol-fog interactions in fog 2? The authors do show aerosol-fog interactions are stronger in Fog 2 than Fog 1 in their table 3. However, the authors don't demonstrate a **causal link** between the increased strength of ACI from Fog 1 to Fog 2 and the ACI in Fog 1. To show conclusively the aerosol-fog interaction is 'self-enhancing' in the simulations as per their own definition, I think the authors would need to show that the aerosols in the first fog affect the aerosol-fog interactions in the second fog. In principle, this could be done with a third simulation, in which the first fog was clean and the second polluted. In this simulation, if the AFIs were weaker in the second fog than in the simulation in which both fogs were polluted, I think the authors' hypothesis would be confirmed.

**Response**: Thank you for your suggestion. We design the third simulation called EXP3, as you suggested. Fog1 is under clean conditions (5% of emission from the MEIC database) and Fog2 is under polluted conditions (the default emission from the MEIC database). Particularly, according to Fog1 dissipate time, the clean condition is set before 11:00 LST on 26 November 2018, and the polluted condition is set after 12:00 LST on 26 November 2018. In Table 4, two fog events in the EXP1 are both under polluted conditions. The EXP2 represents that the two fog events are both under clean conditions. The response of fog optical depth to the change of droplet number concentration ($\Delta\ln FOD/\Delta\ln N_f$), from the EXP2 to the EXP3 is 1.17 in Fog2, smaller than 1.32 from the EXP2 to the EXP1. Therefore, the aerosol–fog interaction (AFI) in Fog1 can affect AFI in Fog2.

We have revised the simulation design accordingly (Page 7, Lines 160-166) and the analysis accordingly (Page 10, Lines 231-233).

**Table 4.** Quantitative estimation of AFI strength in two fog events (Fog1 and Fog2), including the responses of fog optical depth (FOD), liquid water path (LWP), and fog effective radius ($R_e$) to the changes in fog droplet number concentration ($N_f$). The EXP1 is that two fog events are both under polluted conditions, and EXP2 is under clean conditions. The EXP3 is that Fog1 is under clean conditions and Fog2 is under polluted conditions. The ratio is the relative change between Fog1 and Fog2, calculated as (Fog2 − Fog1)/Fog1. In the fourth and sixth columns, Fog1 in both EXP2 and EXP3 is under clean conditions.

| | EXP1 vs EXP2 | | | EXP3 vs EXP2 | | |
|---|---|---|---|---|---|---|
| | Fog1 | Fog2 | Ratio | Fog1 | Fog2 | Ratio |
| $\Delta \ln FOD/\Delta \ln N_f$ | 0.98 | 1.32 | 34.7% | – | 1.17 | – |
| $\Delta \ln LWP/\Delta \ln N_f$ | 0.76 | 1.08 | 42.1% | – | 1.00 | – |
| $-\Delta \ln R_e/\Delta \ln N_f$ | 0.22 | 0.24 | 9.1% | – | 0.17 | – |

4. The authors also need to show how the absolute PM2.5 concentration varies with time through the two fog events (preferably both in simulations and observations). Otherwise, the results in Table 3 are not useful, as the AFIs might get stronger simply because aerosol concentrations get higher. Furthermore, for the same reason, it would be useful to show the timeseries of aerosol number concentrations (perhaps > 100nm diameter) in the two simulations.

**Reply**: The timeseries of PM$_{2.5}$ mass concentration and aerosol number concentration are shown in Fig. S4. PM$_{2.5}$ mass concentration is similar before Fog1 and Fog2

formation, and aerosol number concentration before Fog2 is less than that before Fog1 formation. Therefore, changes in aerosol concentration are not the main reason for increasing aerosol-induced changes in the two fog properties. The above discussions are added (Pages 9-10, Lines 219-223), and Figure S4 is added in the supplement.

[Figure]

**Figure S4.** The timeseries of $PM_{2.5}$ mass concentration and aerosol number concentration in Nanjing (the blue line: observed $PM_{2.5}$ mass concentration, the red line: simulated $PM_{2.5}$ mass concentration, and the green line: simulated $PM_{2.5}$ number concentration). Fog1 and Fog2 in the light grey box are the two fog events. Time '2512' indicates 12:00 local standard time (LST) (LST = Universal Time Coordinated + 8 h) on 25 November 2018. The other time expressions follow the same logic.

5. Figure 4 is very hard to interpret quantitatively. Is LWP from Himawari available as it is, for example, from MODIS, GOES or SEVIRI? Could it be used instead of the visible light images?

**Reply**: LWP is not available in Himawari products, but COD is available. The monitoring time of MODIS satellite is too late because fog events have dissipated. The

monitoring range of geostationary satellites GOES and SEVIRI cannot cover the fog area in our article. Therefore, we use the COD products to replace the visible light images in Fig. 4.

We revised the sentences (Pages 8, Lines 181-187): "Figure 4 shows the evaluation of fog spatial distribution. The simulated fog optical depth (FOD) distribution is compared with the Himawari-8 cloud optical depth products and ground-based observations (the black circles in Fig. 4) at 08:00 LST on 26 and 27 November 2018, respectively. Qualitatively, the simulated fog spatial distribution and magnitude are generally consistent with satellite and ground-based observations. Similarly, Lee et al. (2016) evaluated fog distribution simulation against cloud optical depth from satellite; they also concluded that the distributions of simulation and observation were generally comparable with each other."

[Figure]

Figure 4. (a, c) Distributions of ground-based fog observations (the circular points) and cloud optical depth from Himawari-8 products at 08:00 LST on 26-27 November 2018. (b, d) Simulated fog optical depth (FOD) distributions in the domain 03 at the

corresponding time of observations. Time '2608LST' indicates 08:00 local standard time (LST) (LST = Universal Time Coordinated + 8 h) on 26 November 2018. The other time expressions follow the same logic.

**Minor Comments**

1. In the abstract the authors say "AFIs in the first fog…result in higher [droplet] number concentration …in Fog 2 than in Fog 1. For this to be true, my first thought was that AFIs in the first fog would have to reduce scavenging of aerosol and result in higher aerosol concentration in the second fog than would have been the case if the first fog hadn't formed. The authors don't show this. They do show that Fog 1 changes meteorological conditions, which might indirectly affect droplet concentration in Fog 2 by changing LWC in Fog 2, but starting the list at line 21 with droplet concentration rather than LWC implies (to me at least) that the main mechanism is an aerosol one: aerosol-fog interaction in Fog 1 affect aerosols in Fog 2, which then change droplet concentration in Fog 2, which then changes LWC and lifetime (the classic ACI pathway). The authors don't have any evidence for that (the mechanism is meteorology, not aerosols).

**Reply**: Sorry for the misleading sentences. We agree with the reviewer that the mechanism is meteorology, not aerosols. Therefore, we have revised the abstract (Page 1, Lines 20-24): "Our simulations indicate that conducive PBL conditions are affected by AFI with high aerosol loading in Fog1, and then PBL promotes AFI in Fog2, resulting in higher liquid water content, higher droplet number concentration, smaller droplet size, larger fog optical depth, wider fog distribution, and longer fog lifetime in Fog2 than in Fog1."

2. Line 45 "proven" – I would say "showed" – a 'pivotal role' is not a mathematical concept so it is not really 'proved'.

**Reply**: We have revised the sentence accordingly (Page 2, Lines 49-50): "The critical roles of aerosols and the planetary boundary layer (PBL) in these processes have been shown (Boutle et al., 2018; Niu et al., 2011; Quan et al., 2021). "

3.   Line 70 – what is the 'critical turbulence coefficient'? The reader should not need to look up the literature unless they are very unfamiliar with fog.

**Reply**: "The critical turbulence coefficient was the turbulence threshold for diagnosing whether turbulence suppressed fog or not. If the turbulence intensity inside fog was weaker than the critical turbulence coefficient, the fog persisted; otherwise, the fog dissipated (Zhou and Ferrier, 2008)." The above description is added (Page 4, Lines 81-84).

4.   Line 115  -the innermost simulation still has quite coarse spatial resolution. How well can this resolve the turbulence? Is there a sub-grid cloud parameterization in the model, or does the Grell 3D cumulus scheme lead to sub-grid variability in fog?

**Reply**: Turbulence is parameterized in the planetary boundary layer scheme. Based on closure theory, turbulent fluxes are calculated from gradients and parameterized vertical mixing coefficients. Besides, the parameterized vertical mixing coefficient also affects the vertical distribution of meteorological elements by the heat diffusion equation. Considering the consumption of computing cost, this kind of planetary boundary layer scheme is widely used in mesoscale numerical models (such as WRF), though its accuracy is not as good as the large eddy simulation.

There is a sub-grid cloud parameterization in the MYNN2.5 planetary boundary layer scheme, instead of the Morrision microphysics scheme or the Grell 3D cumulus scheme. The sub-grid cloud parameterization can be found in the reference paper (Chaboureau and Bechtold, 2002), which is consistent with the source code in MYNN2.5 planetary boundary layer scheme. The sub-grid cloud water content is

derived from a function of the normalized saturation deficit. So, sub-grid cloud parameterization is considered in our paper.

We revised the sentences (Page 6, Line 144-145): "Turbulence is parameterised in the MYNN2.5 scheme and there is also a sub-grid cloud parameterisation (Chaboureau and Bechtold, 2002) in the MYNN2.5 scheme."

5. Line 165- what would be a perfect HSS score? Is the score calculated using each gridbox as input? Please be clearer about how this evaluation was done.

**Reply**: "A perfect HSS score is 1.0, indicating that simulations are identical to observations. We used the fog occurrence at the observation stations and the closest model grids as input for HSS score. In our study, the HSS score are 0.34 and 0.36 in Fog1 and Fog2, respectively, which are close to previous reports (Mecikalski et al., 2008; Xu et al., 2020; Yamane et al., 2010)." The above description is added in the revised manuscript (Pages 8-9, Lines 196-200).

6. Figure 9: Is 'fog optical depth per unit height' the same as "average extinction coefficient through the fog"? It might help the reader to explain this in the caption.

**Reply**: Yes, 'fog optical depth per unit height' is the same as "average extinction coefficient through the fog". We have revised the sentences in the revised manuscript (Page 44, lines 872-873; Page 14, Lines 325, 332).

**References**

Boutle, I., Price, J., Kudzotsa, I., Kokkola, H., and Romakkaniemi, S.: Aerosol–fog interaction and the transition to well-mixed radiation fog, Atmos. Chem. Phys., 18, 7827-7840, https://doi.org/10.5194/acp-18-7827-2018, 2018.

Chaboureau, J.-P. and Bechtold, P.: A Simple Cloud Parameterization Derived from Cloud Resolving Model Data: Diagnostic and Prognostic Applications, J. Atmos. Sci., 59, 2362-2372, https://doi.org/10.1175/1520-0469(2002)059<2362:ascpdf>2.0.co;2, 2002.

Lee, H.-H., Chen, S.-H., Kleeman, M. J., Zhang, H., DeNero, S. P., and Joe, D. K.: Implementation of warm-cloud processes in a source-oriented WRF/Chem model to study the effect of aerosol mixing state on fog formation in the Central Valley of California, Atmos. Chem. Phys., 16, 8353-8374, https://doi.org/10.5194/acp-16-8353-2016, 2016.

Mecikalski, J. R., Bedka, K. M., Paech, S. J., and Litten, L. A.: A Statistical Evaluation of GOES Cloud-Top Properties for Nowcasting Convective Initiation, Mon. Weather Rev., 136, 4899-4914, https://doi.org/10.1175/2008mwr2352.1, 2008.

Niu, S. J., Liu, D. Y., Zhao, L. J., Lu, C. S., Lü, J. J., and Yang, J.: Summary of a 4-Year Fog Field Study in Northern Nanjing, Part 2: Fog Microphysics, Pure Appl. Geophys., 169, 1137-1155, https://doi.org/10.1007/s00024-011-0344-9, 2011.

Quan, J., Liu, Y., Jia, X., Liu, L., Dou, Y., Xin, J., and Seinfeld, J. H.: Anthropogenic aerosols prolong fog lifetime in China, Environ. Res. Lett., 16, 044048, https://doi.org/10.1088/1748-9326/abef32, 2021.

Xu, X., Lu, C., Liu, Y., Gao, W., Wang, Y., Cheng, Y., Luo, S., and Van Weverberg, K.: Effects of Cloud Liquid-Phase Microphysical Processes in Mixed-Phase Cumuli Over the Tibetan Plateau, J. Geophys. Res.: Atmos., 125, https://doi.org/10.1029/2020jd033371, 2020.

Yamane, Y., Hayashi, T., Dewan, A. M., and Akter, F.: Severe local convective storms in Bangladesh: Part II, Atmos. Res., 95, 407-418, https://doi.org/10.1016/j.atmosres.2009.11.003, 2010.

Yan, S., Zhu, B., Zhu, T., Shi, C., Liu, D., Kang, H., Lu, W., and Lu, C.: The Effect of Aerosols on Fog Lifetime: Observational Evidence and Model Simulations, Geophys. Res. Lett., 48, https://doi.org/10.1029/2020gl091156, 2021.

Zhou, B. and Ferrier, B. S.: Asymptotic Analysis of Equilibrium in Radiation Fog, J. Appl. Meteorol. Clim., 47, 1704-1722, https://doi.org/10.1175/2007jamc1685.1, 2008.

---

## Author Comment (AC3)

Dear Professor Graham Feingold,

Thank you for your constructive comments. Briefly, we have addressed all your comments. The corresponding responses are listed below. Furthermore, the Wiley Editing Services (https://editingservices.wiley.cn/) provide thorough English language editing.

With regards,

Naifu Shao, Chunsong Lu*, and co-authors.

**Comments:**

I note that nowhere in your manuscript do you discuss the role of shortwave radiation and its affect on fog lifetime. This is an important part of the discussion. With increasing liquid water path (LWP) and increasing drop concentration, SW heating increases. There are also longwave aerosol-related effects: at low LWP ($<\sim 25$ g/m2); an increase in drop concentration will increase radiative cooling. I also note that there is no quantitative information on LWP, only its response (e.g., Table 4). This is really important information if one is to understand the radiation interactions.

**Response**: Thank you for your suggestion. We agree that it is important to discuss the role of shortwave radiation during fog dissipation and its effects on fog properties. We have revised and reorganized the related sentences.

Page 12, Lines 282-285: Compared with clean conditions, the larger $\tau_t$ (mainly due to larger $\tau_c$) and delayed fog dissipation in polluted conditions reduce short-wave radiation reaching the ground (from $-46$ W m$^{-2}$ to $-121$ W m$^{-2}$) during the Fog1 dissipation time, leading to a decrease in T$_{2m}$ (from $-0.2$ °C to $-1$ °C) and PBLH (from $-42$ m to $-118$ m) (Fig. 7).

Page 14, Lines 325-330: "As shown in Fig. S5, LWP is larger under polluted conditions than that under clean conditions, particularly for Fog2. The average LWP in Fog1 and

Fog2 under polluted conditions are 11.6 g m$^{-2}$ and 24.3 g m$^{-2}$, respectively. Therefore, $\tau_c$ in Fog2 (4.9) is larger than that in Fog1 (2.1). Similar to previous studies (Jiang et al., 2001; Williams and Igel, 2021), higher LWP or $\tau_c$ leads to stronger long-wave radiative cooling at the fog-top."

**A linguistic revision**

1)  What are "conducive PBL conditions"? I think you mean "conducive to fog formation". Please make changes throughout.

**Reply**: Yes, we mean "conducive to fog formation". We have modified the phrase to be: "PBL conditions conducive to Fog2 formation". We have revised the entire article.

2)  change 'scenario' to 'event'

**Reply**: All 'scenario' in our article have been revised to 'event'.

3)  remove all "the" before "EXPn"

**Reply**: All "the" before "EXPn" has been removed.

4)  dissipation time (not dissipate time)

**Reply**: All "dissipate time" has been revised to "dissipation time".

5)  Fog 1 occurs under clean conditions (not Fog1 161 is under the clean condition)

**Reply**: We have revised "is under the clean conditions" to "occurs under clean conditions".

6)  "under clean and polluted conditions" (remove 'the')

**Reply**: It has been removed in our article.

7)  What is 'more remarkable AFI'? Do you mean stronger AFI? Please be clear.

**Reply**: Yes, 'more remarkable AFI' means stronger AFI. We have revised it in our article.

8) "can enhance cooling" do you mean "enhances cooling"? Please use clear causal language if that's what you mean. There are many instances "can affect", "can indicate".

**Reply**: Yes, we mean that "enhances cooling". We have revised all the related phrases.

9) You have a tendency to create new acronyms like AFI, FOD, TOD, which makes the manuscript less readable to the broader audience. The use of symbols significantly alleviates this problem (e.g., \tau_f, \tau_t). Even AFI might be unnecessary given the familiar ACI. (You could simply point out that ACI in fog has its own particular questions). Also, why N_f when N_d (drop concentration) or N_c (cloud droplet concetration) are widely used. And \tau_c would be better than \tau_f. As it is you use other standard cloud-related acronyms such as LWP, LWC.

**Reply**: Thank you for your suggestions to make the manuscript more readable to the broader audience. We have revised the acronyms and symbols accordingly. AFI, FOD, TOD and $N_f$ have been revised to ACI, $\tau_c$, $\tau_t$ and $N_d$, respectively.

**References**

Jiang, H., Feingold, G., Cotton, W. R., and Duynkerke, P. G.: Large-eddy simulations of entrainment of cloud condensation nuclei into the Arctic boundary layer: May 18, 1998, FIRE/SHEBA case study, J. Geophys. Res.: Atmos., 106, 15113-15122, 10.1029/2000jd900303, 2001.

Williams, A. S. and Igel, A. L.: Cloud Top Radiative Cooling Rate Drives Non-Precipitating Stratiform Cloud Responses to Aerosol Concentration, Geophys. Res. Lett., 48, 10.1029/2021gl094740, 2021.

---

## Author Comment (AC4)

**Response to Referee #1**

Dear Referee,

We appreciate your positive and constructive comments. We have read these comments carefully and made revisions accordingly. The responses to the comments are listed below.

Sincerely,

Naifu Shao, Chunsong Lu*, and all co-authors.

**Main comments:**

This manuscript presents a modelling study of two successive fog events in the Yantze River Delta region of China. It aims to show how the fog properties in the second event are influenced by the first. I find this a truly interesting topic and exciting approach. However, I struggled with the manuscript for the following reasons:

**Response**: Thank you for your valuable comments.

1.  The central message is that fog properties are influenced by aerosol as well as other boundary-layer conditions. The latter may be modified by a preceding fog event, resulting in fog property differences between both events. (a) This simple -- and very interesting – finding is hidden behind the phrase "self-enhanced AFIs", and thus took me more time to understand than would have been necessary. I would suggest to focus on the changes to the fog rather than "AFIs", and to speak about "aerosol loading" or "polluted conditions" to clarify the meteorological context. (b) Also, "AFIs", which is modelled on the common abbreviation "ACI" for aerosol-cloud interactions should probably lose the "s" to make it consistent with ACI. (c) A change of the title could also

be considered to more close reflect the paper's focus, e.g. "Radiation fog properties in two consecutive events under clean and polluted conditions..." or similar.

**Response**: Thank you for your suggestion.

(a) We agree with the referee and have deleted the phrase "Self-enhanced AFI" in many places of the abstract and main text (e.g., Page 1 Line 25; Page 5 Lines 114-116). Instead, we focus more on the changes to the fog, as suggested by the referee. We also speak about "aerosol loading" or "polluted conditions" to clarify the meteorological context. For example:

- "Our simulations indicate that the PBL conditions conducive to Fog2 formation are affected by ACI with high aerosol loading in Fog1; subsequently, PBL promotes ACI in Fog2, resulting in a higher liquid water content, higher droplet number concentration, smaller droplet size, larger fog optical depth, wider fog distribution, and longer fog lifetime in Fog2 than in Fog1." (Page 1, Lines 21-25).

- "The two fog events provide an excellent opportunity to investigate ACI under polluted conditions as a chain. This involves analysing how high aerosol loading affects properties in the first fog event, how the properties in the first polluted fog event affect radiation and PBL structure, and finally, how radiation and PBL affect properties and ACI in the second fog event under polluted conditions" (Page 5, Lines 104-108).

- "Here, we study how radiation fog properties are affected by high aerosol loading and PBL meteorological conditions in two successive events in the YRD region" (Page 5, Lines 118-119).

- "Furthermore, compared with the difference in aerosol-induced changes in $RH_{2m}$ and PBLH before fog formation, $RH_{2m}$ increases by 6% and PBLH decreases by 92 m under polluted conditions, which is larger than those ($RH_{2m}$: 4% and PBLH: −59 m) under clean conditions" (Page 12, Lines 271-274).

(b) All the "AFIs" in this manuscript are revised to "ACI".

(c) According to your suggestion, we have revised the title to be "Radiation fog properties in two consecutive events under polluted and clean conditions in the Yangtze River Delta, China: A simulation study".

2. (a)The state of the art chapter does not seem complete. The central motivation, i.e. limited knowledge about AFI, is only briefly stated, and not explained (line 81). (b) The fundamental premise that an event may be influenced by a previous event does not follow from the literature review presented at all. (c) The focus, concepts and terminology of the first research question are neither derived from the literature, nor are they explained. (1) What is a "stronger" fog scenario? (2) What does "stronger AFIs" mean? (3) What would you expect? Why? And why does it matter?

**Response**: Thank you for your suggestion.

(a) Regarding limited knowledge about aerosol–cloud interaction (ACI) in fog, we meant that it is not clear how ACI and planetary boundary layer (PBL) interacts with each other and the evolution of ACI in successive fog scenarios remains unknown. To make the description clearer, we pointed out the questions directly (Page 4, Lines 98-101): "What are the physical mechanisms behind the property changes during two successive fog events? Furthermore, which fog event has macro- and microphysical properties that are more sensitive to aerosol loading, i.e., experiencing a stronger ACI? What are the mechanisms underlying the interactions between ACI and PBL?"

(b) We have added the sentences to show that an event may be influenced by a previous event (Page 4, Lines 95-98): "Previous studies typically focused on either a single fog event or analysed multiple fog events statistically; however, several studies noted that LWC, $N_d$, and liquid water path (LWP) in a latter fog event exhibited larger values compared to those for the preceding fog event (Quan et al., 2011; Wærsted et al., 2017)."

(c) The focus, concepts and terminology are explained as follows.

(1) "Stronger fog scenario" means a fog event has larger macro- and microphysical

properties, such as fog droplet number concentration and liquid water content. To be more specific, we have improved the description (Page 4, Lines 96-97): "liquid water content, droplet number concentration and liquid water path in a latter fog event exhibited larger values compared to those for the preceding fog event."

(2) AFI is replaced by ACI. "Stronger ACI" means the more remarkable fog property response to changes in aerosol loading. For example, if aerosol-induced changes in fog optical depth is larger, ACI in fog is stronger. We have added the above explanation (Page 4, Lines 96-98): "which fog event has fog macro- and microphysical properties that are more sensitive to aerosol loading, i.e., experiencing stronger ACI?"

(3) The reason to analyze the evolution of ACI in two fog scenarios is that stronger ACI can affect fog development, for example, increasing droplet number concentration more significantly. Furthermore, we would like to examine the mechanisms responsible for the evolution of ACI and study how the interaction between ACI and PBL make fog properties change in the two successive fog events. We have revised the manuscript accordingly:

- "However, it is not clear how ACI in the first fog (Fog1) affects PBL, and subsequently affects ACI in the second fog (Fog2), which is important to understand the interaction between ACI and PBL as well as their effects on fog properties" (Page 1, Lines 16-19).

- "Our simulations indicate that the PBL conditions conducive to Fog2 formation are affected by ACI with high aerosol loading in Fog1; subsequently, PBL promotes ACI in Fog2, resulting in a higher liquid water content, higher droplet number concentration, smaller droplet size, larger fog optical depth, wider fog distribution, and longer fog lifetime in Fog2 than in Fog1" (Page 1, Lines 21-25).

3. In some places, aspects concerning methodology and interpretation remain unclear. (a)How precisely is the validation performed? (b)To what extent and under what

conditions can the findings of this study be generalized? (c)Instead of using "AFIs", in many places it would be more helpful to explicitly address the parameter of relevance, e.g. LWP, aerosol loading, droplet radius...

**Response**: Thank you for your suggestion.

(a)  We add Table 3 to explain the elements a–d in the Heidke skill score (HSS). In our study, the HSS score are 0.34 and 0.36 in Fog1 and Fog2, respectively, which are close to previous reports (Mecikalski et al., 2008; Xu et al., 2020; Yamane et al., 2010). We have added the above description (Page 9, Lines 203-205).

**Table 3.** Elements a–d in the Heidke skill score calculation

|                 | Fog observed | No fog observed |
| --------------- | ------------ | --------------- |
| Fog simulated   | a            | b               |
| No fog simulated | c           | d               |

(b)  "Our findings are generalised for the following reasons: First, the simulation design is reasonable. According to previous simulation studies, polluted and clean conditions are simulated by varying emission intensity. Second, the conclusions are robust, because they are derived from physical analyses. The interactions between aerosol loading, fog macro- and microphysical properties, and boundary layer meteorological conditions are understood physically. Third, fog events are typical and have large coverage. Therefore, the findings of this study are generalisable, at least for polluted fog events during winter." The above discussions are added (Page 20, Lines 457-463).

(c)  Thank you for your suggestion. AFI is replaced by parameters of relevance.

●  "During the daytime before Fog2 formation, meteorological conditions in the PBL are affected by $\tau_c$ at the Fog1 dissipation stage. Compared with clean conditions, the larger $\tau_t$ (mainly due to larger $\tau_c$) and delayed fog dissipation in polluted conditions reduce short-wave radiation reaching the ground (from $-46$ W m$^{-2}$ to $-121$ W m$^{-2}$) during the Fog1 dissipation time, leading to a decrease in $T_{2m}$ (from $-0.2$ °C to $-1$ °C) and PBLH (from $-42$ m to $-118$ m) (Fig. 7)" (Page 12, Lines 280-285).

- "Larger $\tau_c$ and delayed dissipation result in lower temperature, higher relative humidity, and higher stability by affecting solar radiation during the daytime" (Page 13, Lines 301-302).

- "The cold centre is related to lower temperature under polluted conditions due to larger $\tau_c$ and longer duration in Fog1" (Page 13, Lines 306-307).

4.  While the paper is both legible and intelligible, it would profit from a linguistic revision.

Response: Thank you for your comment. We have revised the sentences according to Editor's suggestion. We also hope this manuscript has been improved after a linguistic revision supported by Wiley Editing Services (https://editingservices.wiley.cn/).

**DETAILS**

(1) 15 - "pivotal" is unclear here

**Reply**: The word "pivotal" is replaced by "critical" (Page 1, Line 16).

(2) 15 - what is "the fog cycle"?

**Reply**: We mean the fog life cycle. We have revised the sentence: "Aerosol–cloud interaction (ACI) in fog and planetary boundary layer (PBL) conditions play critical roles in the fog life cycle" (Page 1, Lines 15-16).

(3)  16: Why should they focus on these differences? What is special about successive events?

(4) 17: What knowledge gap exactly?

**Reply**: We would like to reply to the two comments together, because they are closely related to each other. The difference between two successive events is important to understand the interaction between Aerosol–cloud interaction (ACI) and planetary boundary layer (PBL) as well as their effects on fog properties. That is why we are interested in the difference between two fog events. However, it is not clear how ACI in the first fog affects PBL and then ACI in the second fog. This is the knowledge gap. We have revised the abstract accordingly (Page 1, Lines 16-19).

(5) 19: "AFIs ... promote..." -- Do you mean high/low aerosol loadings? Or the interaction (mechanisms) specifically?

**Reply**: We mean the interaction (mechanisms) specifically. We have revised the sentence (Page 1, Lines 21-25): "Our simulations indicate that the PBL conditions conducive to Fog2 formation are affected by ACI with high aerosol loading in Fog1; subsequently, PBL promotes ACI in Fog2, resulting in a higher liquid water content, higher droplet number concentration, smaller droplet size, larger fog optical depth, wider fog distribution, and longer fog lifetime in Fog2 than in Fog1."

(6) 22: "is defined as" -- you mean that you define it as, or is this taken from elsewhere?

**Reply**: We mean that we define it as. This phrase is deleted because self-enhanced AFI is deleted, according to the referee's other comments.

(7) 38: fog does not lead "to environmental pollution" - please clarify this statement

**Reply**: We agree with the referee and have deleted this phrase (Page 2, Lines 41-43): "This results in low visibility, affecting human health, transportation, and power system (Niu et al., 2010)."

(8) 40: You state that the "physical processes of fog remain unclear". What exactly do you refer to? Can you provide a reference, please? I would think that the processes are pretty well understood.

Reply: We have reorganized the sentences to describe the unclear physical processes of fog and have added references (Page 2, Lines 43-47): "An important reason is that the physical processes of fog remain unclear because many processes (aerosol activation, condensation, radiation, and turbulence) occur simultaneously and interact with each other nonlinearly (Haeffelin et al., 2010), which affects fog properties (Mazoyer et al., 2022) and impedes related parameterisation (Poku et al., 2021)."

(9) 47: First sentence is a repetition of statement in line 36.

Reply: We have revised the sentence (Page 3, Lines 54-56): "Since fog is a special type of cloud (Guo et al., 2021; Kim and Yum, 2010, 2013; Wang et al., 2023), aerosol–fog interaction is expected to share similarities with aerosol–cloud interaction (ACI)."

(10) 52: What do you mean by "fog number concentration"? droplet number concentration in fog?

Reply: Yes, we mean fog droplet number concentration. The phrase is revised accordingly (Page 3, Line 63).

(11) 53: Can these numbers be generalized? How would they be expected to change given different environmental conditions? Is this continental radiation fog, sea fog, advection fog over land, ...?

Reply: The referee's concern is reasonable. Here we take the two fog field campaigns as examples representing polluted and clean conditions, respectively. Although the field campaign in the North China Plain cannot fully stand for all polluted conditions and the

field campaign in Xishuangbanna, China, cannot fully stand for all clean conditions, the comparison between the two examples does show the difference of fog properties between polluted and clean conditions, i.e., fog droplet number concentration is higher and effective radius is smaller in polluted conditions than in the clean one. Examples above are both continental radiation fog.

We have revised the sentences as follow (Page 3, Lines 60-66): "Various continental fog observation projects showed that fog microphysical properties were significantly affected by aerosol loading (Mazoyer et al., 2019; Niu et al., 2011; Quan et al., 2011; Wang et al., 2021). For instance, in polluted fog observations, Quan et al. (2011) found that the fog droplet number concentration ($N_d$) was higher than 1,000 cm$^{-3}$ and effective radius ($R_e$) was approximately 7 μm in the North China Plain. In clean fog observations, Wang et al. (2021) showed that $N_d$ was smaller than 100 cm$^{-3}$ and $R_e$ was approximately 9 μm in the tropical rainforest in Xishuangbanna, China."

(12) 70: That radiative cooling "is an important factor for temperature inversion, providing stable conditions for fog formation" is not a finding of the cited studies in the 2010s, but can be derived from very basic textbook knowledge.

**Reply**: We agree with the referee and have revised the sentence (Page 4, Lines 79-81): "Previous studies showed that radiative cooling was an important factor in temperature inversion that provided stable conditions for fog formation (Fitzjarrald and Lala, 1989; Holets and Swanson, 1981; Roach et al., 1976)."

(13) 81: In what respect is this knowledge limited? What is lacking?

(14) 83: Why do you think successive fog events are worth considering?

**Reply**: The two comments are replied together. The understanding of ACI in fog remains limited because the mechanism behind the interaction between aerosol, fog and PBL is not fully studied, especially in two successive fog events. We have reorganized

the sentences to make a clearer description (Page 4, Lines 95-101): "Previous studies typically focused on either a single fog event or analysed multiple fog events statistically; however, several studies noted that LWC, $N_{\mathrm{d}}$, and liquid water path (LWP) in a latter fog event exhibited larger values compared to those for the preceding fog event (Quan et al., 2011; Wærsted et al., 2017). What are the physical mechanisms behind the property changes during two successive fog events? Furthermore, which fog event has macro- and microphysical properties that are more sensitive to aerosol loading, i.e., experiencing a stronger ACI? What are the mechanisms underlying the interactions between ACI and PBL?"

(15) 84/5: Why?

**Reply**: The reason is that fog is a special cloud near ground. We have revised the sentence (Page 5, Lines 108-111): "Additionally, since fog is a special type of cloud near the ground, studying the evolution of ACI in fog aids in examining the progression of ACI in cloud, which is critical for climate prediction (Boutle et al., 2018; Vautard et al., 2009)."

(16) 89: How do you define "stronger AFIs", what do you mean by this and why does it matter?

**Reply**: According to editor's comments, AFI is revised to ACI. "Stronger ACI" in fog is defined as the more remarkable fog property response to changes in aerosol concentration. For example, if aerosol-induced change in fog optical depth is larger, ACI in fog is stronger. The reason for comparing ACI strength in two fog scenarios is that stronger ACI promotes fog development, for example, increasing droplet number concentration more significantly. Furthermore, we would like to examine the mechanisms responsible for the evolution of ACI and study how the interaction between ACI and PBL make fog properties change in the two successive fog events. We have revised the manuscript accordingly (Page 4, Lines 99-101, Page 5, Lines 104-108).

(17) 101: What aerosol species?

(18) 101: What is "massive"? Please be more specific.

**Reply**: We would like to reply two comments together. It is $PM_{2.5}$ and the $PM_{2.5}$ mass concentration is over 100 μg m$^{-3}$.

We have revised the sentence (Page 5, Lines 119-121): "Before fog events in the YRD, the $PM_{2.5}$ mass concentration was over 100 μg m$^{-3}$ due to anthropogenic emissions (Zhu et al., 2019)."

(19) 140: What does this experiment consist of, and what sensitivities are tested for?

**Reply**: The control run is tested for polluted conditions with emission intensity directly from the MEIC database. The sensitive experiment is tested for clean conditions with the emission intensity multiplied by 0.05. The design of the control run and sensitivity test is the same as those in Jia et al. (2019) and Yan et al. (2020). We have revised the sentences (Page 7, Lines 159-164).

(20) 160: How do you compare observations and model? How do you define "consistent" in this regard?

**Reply**: We use ground-based fog observations and cloud optical depth from Himawari-8 to evaluate simulations (Fig. 4). "consistent" is defined as similarity of the simulated fog spatial distributions and magnitude of optical depth between simulations and observations. We add "generally" before "consistent" and revised the sentences to be more objective (Page 8, Lines 188-192): "Qualitatively, the spatial distribution and magnitude of the simulated fog are generally consistent with satellite and ground-based observations. Similarly, Lee et al. (2016) evaluated fog distribution simulations against satellite-derived cloud optical depth from satellite and concluded that the distributions

of simulations and observations were generally comparable to each other."

Besides the qualitative evaluation, we also use Heidke skill score (HSS) to quantitatively evaluate the simulations. Please see the response to the next comment.

[Figure]

Figure 4. (a, c) Distributions of ground-based fog observations (the circular points) and cloud optical depth from Himawari-8 products at 08:00 LST on 26 and 27 November 2018. (b, d) Simulated fog optical depth distributions in domain 03 at the corresponding time of observations. Time '2608LST' indicates 08:00 local standard time (LST) (LST = Universal Time Coordinated + 8 h) on 26 November 2018. The other time expressions follow the same logic.

(21) 161: Based on which parameters is HSS calculated?

**Reply**: Table 3 is added to more clearly show how HSS is calculated. The description of HSS score is also revised (Page 9, Lines 196-202): "Elements *a–d* are the numbers of "hits", "false alarms", "misses", and "correct negatives", respectively, which are

determined by observations and simulations as shown in Table 3. To identify observed fog at a station, two criteria are used: visibility less than 1 km and relative humidity larger than 90% (Yan et al., 2020). Simulated foggy grids are classified based on three criteria (Jia et al., 2019; Zhao et al., 2013): fog water mixing ratio over 0.01 g kg$^{-1}$, $N_d$ greater than 1 cm$^{-3}$, and fog base touching the ground. Elements *a–d* are calculated based on the fog occurrence at the observation stations and the closest model grids."

Table 3. Elements a–d in the Heidke skill score calculation

|  | Fog observed | No fog observed |
|---|---|---|
| Fog simulated | a | b |
| No fog simulated | c | d |

(22) 172: Why this threshold?

Reply: "We also test other thresholds, 1%, 2.5%, 7.5%, and 10% (Fig. S3). The results are similar to those based on the threshold of 5%." We have added the above description (Page 9, Lines 211-213). Figure 3 is added in the supplement.

[Figure]

Figure S3. Aerosol effect on relative changes in macro- and microphysical properties

during the first fog (Fog1) and the second fog (Fog2). Figure S3 a–d are the results with fog fraction area thresholds 1%, 2.5%, 7.5%, and 10% respectively. $N_d$, LWC, $R_e$, Area, Height, Duration, LWP, and $\tau_c$ indicate fog number droplet concentration, liquid water content, effective radius, fog area fraction, fog-top height, liquid water path, and fog optical depth, respectively. The ratio of changes is calculated as Polluted/Clean.

(23) Figure 5: I find it slightly confusing that the reference case is shown with 100% bars in all cases. I suggest leaving this out and only showing the polluted (a,b) or fog2 (c) situations.

**Reply**: We agree with the referee and have deleted the 100% bars. As suggested, we only show figure (a, b) and deleted figure c.

[Figure]

**Figure 5.** (a) Aerosol-induced changes in macro- and microphysical properties during the first fog (Fog1) and the second fog (Fog2) events under polluted and clean conditions. (b) Temporal evolution of fog area fraction under clean and polluted conditions. $N_d$, LWC, $R_e$, Area, Height, Duration, LWP, and $\tau_c$ indicate fog droplet number concentration, liquid water content, effective radius, fog area fraction, fog-top height, liquid water path, and fog optical depth, respectively. The ratios of changes are calculated by Polluted/Clean in Fig. 5a which reveal the aerosol-induced changes. The numbers above the bars in Fig. 5a represent the difference in those ratios of changes between Fog1 and Fog2 (calculated by Fog2–Fog1). Time '2522' in Fig. 5b indicates

22:00 local standard time (LST) (LST = Universal Time Coordinated + 8 h) on 25 November 2018. The other time expressions follow the same logic.

(24) 185ff: Here, and in several other places, you assume that AFI lead to changes in fog2. In section 5 you state that fog2 is different because boundary-layer conditions are different after a previous fog event, and not specifically because of the aerosol. Please make sure your reasoning is consistent.

**Reply**: Sorry for the confusion. Fog2 formation is related to the PBL conditions which are affected by ACI. The reasoning is that ACI under polluted conditions postpones the dissipation of Fog1 owing to these two feedbacks and generates PBL meteorological conditions that are more conducive to the formation of Fog2 than those prior to Fog1. These conditions promote the earlier formation of Fog2, further enhancing the two feedbacks and strengthening the ACI in Fog2. We have revised the whole manuscript (e.g., Page 11, Lines 252-253; Page 2, Lines 34-37).

(25) 241: higher stability

**Reply**: Stronger stability has been replaced by higher stability, according to your advice.

We have revised the sentence (Page 13, Line 299): "Therefore, lower temperature, higher relative humidity, and higher stability result from ACI in Fog1 under polluted conditions, contributing to the earlier formation of Fog2."

**References**

[revised manuscript text omitted]

---

## Author Comment (AC5)

**Response to Referee #2**

This paper describes a case study of two fog events on two consecutive days in Nanjing, simulated by WRF-chem, and proposes that aerosol-fog interactions in the first fog promote aerosol-fog interactions in the second. Most of the hypothesis is reasonable: the first fog influences boundary layer turbulence for the second fog, and that influence is affected by aerosols. However, I am not yet convinced whether the hypothesis that *aerosol-fog interaction* in the second fog is affected by aerosol-fog interaction in the first is adequately demonstrated by the simulations in the paper.

Despite this, the paper describes a useful and interesting study of aerosol-fog interactions, which in itself is well worth publishing in ACP. It is also well structured and well written, in general. I recommend that the authors either perform additional simulations to test their hypothesis, they weaken their definition of self-enhancement, or they change the message of the paper to simply highlight aerosol-fog interactions in Nanjing. Either way, in my assessment the article needs major revisions, but assuming the major comments can be addressed, it would be suitable for ACP.

Dear Referee,

Thank you for your positive and constructive comments. We have addressed your comments and the corresponding replies are listed below. Briefly, we have performed additional simulations, weakened the definition of self-enhancement and changed the message of the paper to simply highlight aerosol-fog interactions in Nanjing, according to your suggestions.

With regards,

Naifu Shao, Chunsong Lu*, and co-authors.

**Major comments**

1.   The authors' summary of their evidence for their hypothesis of 'self-enhanced aerosol-fog interactions' is that by increasing droplet concentrations and by postponing the dissipation of the first fog and promoting the earlier formation of the second, aerosols increase the fog thickness and prolong its lifetime.

**Reply**: Yes, we agree with you. "ACI under polluted conditions postpones the dissipation of Fog1 owing to these two feedbacks and generates PBL meteorological conditions that are more conducive to the formation of Fog2 than those prior to Fog1. These conditions promote the earlier formation of Fog2, further enhancing the two feedbacks and strengthening the ACI in Fog2" (Page 2, Line 34-37).

2.   Figure 7 shows the meteorological differences that arise during the first fog between clean and polluted conditions persist into the second fog. This figure is key. But would these meteorological differences still persist if the second fog, and the period between the fogs, were not also polluted? Can the authors demonstrate that direct aerosol-meteorology interactions during the clear-sky period between the two fogs do not lead to the meteorological differences in Figure 7 and the early onset of Fog 2?

**Response**: Thank you for your suggestion. We think it is possible to reply to this comment by examining the meteorological conditions before Fog 1, instead of examining the conditions before Fog 2. The reason is that there is no fog before Fog 1; all the differences of meteorological conditions before Fog 1 is caused by aerosol-meteorology interaction, which was the question the reviewer asked. "As shown in Table 5, $RH_{2m}$ and PBLH before Fog1 on 25 November under clean conditions are 76% and 669 m, respectively, similar to those under polluted conditions (76% and 670 m, respectively). Therefore, it is unlikely that aerosol-meteorology interaction leads to the meteorological differences in Fig. 7. In addition, a previous study (Yan et al., 2021) also noted that aerosol–fog interaction was more remarkable than aerosol–radiation

interaction." The above discussions are added in the revised manuscript (Page 13, Lines 293-298).

3.   Assuming the authors can demonstrate this, their theory is aerosol-fog interaction in Fog 1 changes meteorology which enhances aerosol-fog interaction in Fog 2. They show the first part of this in Figure 7: aerosol-fog interactions affect meteorology. It's reasonable that this influences the formation time of Fog 2 in the simulations. But does it also influence aerosol-fog interactions in fog 2? The authors do show aerosol-fog interactions are stronger in Fog 2 than Fog 1 in their table 3. However, the authors don't demonstrate a **causal link** between the increased strength of ACI from Fog 1 to Fog 2 and the ACI in Fog 1. To show conclusively the aerosol-fog interaction is 'self-enhancing' in the simulations as per their own definition, I think the authors would need to show that the aerosols in the first fog affect the aerosol-fog interactions in the second fog. In principle, this could be done with a third simulation, in which the first fog was clean and the second polluted. In this simulation, if the AFIs were weaker in the second fog than in the simulation in which both fogs were polluted, I think the authors' hypothesis would be confirmed.

**Response**: Thank you for your suggestion. We design the third simulation called EXP3, as you suggested. Fog1 occurs under clean conditions (5% of emission from the MEIC database) and Fog2 occurs under polluted conditions (the default emission from the MEIC database). In particularly, according to Fog1 dissipation time, clean condition are set before 11:00 LST on 26 November 2018 and polluted conditions are set after 12:00 LST on 26 November 2018. In Table 4, two fog events in EXP1 are both under polluted conditions. EXP2 represents that the two fog events are both under clean conditions. The response of fog optical depth to the change of droplet number concentration ($\Delta\ln\tau_c/\Delta\ln N_d$), from EXP2 to EXP3 is 1.17 in Fog2, smaller than 1.32 from EXP2 to EXP1. Therefore, the aerosol–cloud interaction (ACI) in Fog1 affects ACI in Fog2.

We have revised the simulation design accordingly (Page 7, Lines 164-168) and the analysis accordingly (Page 10, Lines 236-239).

**Table 4.** Quantitative estimation of ACI strength in two fog events (Fog1 and Fog2), including the responses of fog optical depth ($\tau_c$), liquid water path (LWP), and fog effective radius ($R_e$) to the changes in fog droplet number concentration ($N_d$). In EXP1, both fog events occur under polluted conditions, and fog events in EXP2 occur under clean conditions. In EXP3, Fog1 occurs under clean conditions and Fog2 occurs under polluted conditions. The ratio represents the relative change between Fog1 and Fog2, calculated as (Fog2 − Fog1)/Fog1. In the fourth and sixth columns, Fog1 occurs under clean conditions in both EXP2 and EXP3.

| | EXP1 vs EXP2 | | | EXP3 vs EXP2 | | |
|---|---|---|---|---|---|---|
| | Fog1 | Fog2 | Ratio | Fog1 | Fog2 | Ratio |
| $\Delta\ln\tau_c/\Delta\ln N_d$ | 0.98 | 1.32 | 34.7% | _ | 1.17 | _ |
| $\Delta\ln LWP/\Delta\ln N_d$ | 0.76 | 1.08 | 42.1% | – | 1.00 | – |
| $-\Delta\ln R_e/\Delta\ln N_d$ | 0.22 | 0.24 | 9.1% | – | 0.17 | – |

4. The authors also need to show how the absolute PM2.5 concentration varies with time through the two fog events (preferably both in simulations and observations). Otherwise, the results in Table 3 are not useful, as the AFIs might get stronger simply because aerosol concentrations get higher. Furthermore, for the same reason, it would be useful to show the timeseries of aerosol number concentrations (perhaps > 100nm diameter) in the two simulations.

**Reply**: The timeseries of PM$_{2.5}$ mass concentration and aerosol number concentration are shown in Fig. S4. PM$_{2.5}$ mass concentration is similar before Fog1 and Fog2

formation, and aerosol number concentration before Fog2 is less than that before Fog1 formation. Therefore, changes in aerosol concentration are not the main reason for the increase in aerosol-induced changes in the two fog properties. The above discussions are added (Pages 10, Lines 235-239), and Figure S4 is added in the supplement.

[Figure]

**Figure S4.** The timeseries of PM$_{2.5}$ mass concentration and aerosol number concentration in Nanjing (the blue line: observed PM$_{2.5}$ mass concentration, the red line: simulated PM$_{2.5}$ mass concentration, and the green line: simulated PM$_{2.5}$ number concentration). Fog1 and Fog2 in the light grey box are the two fog events. Time '2512' indicates 12:00 local standard time (LST) (LST = Universal Time Coordinated + 8 h) on 25 November 2018. The other time expressions follow the same logic.

5. Figure 4 is very hard to interpret quantitatively. Is LWP from Himawari available as it is, for example, from MODIS, GOES or SEVIRI? Could it be used instead of the visible light images?

**Reply**: LWP is not available in Himawari products, but COD is available. The monitoring time of MODIS satellite is too late because fog events have dissipated. The monitoring range of geostationary satellites GOES and SEVIRI cannot cover the fog

area in our study. Therefore, we use the COD products to replace the visible light images in Fig. 4.

We revised the sentences (Page 8, Lines 185-192): "Figure 4 shows the evaluation of fog spatial distribution. The simulated fog optical depth distribution is compared with the Himawari-8 cloud optical depth products and ground-based observations (black circles in Fig. 4) at 08:00 LST on 26 and 27 November 2018, respectively. Qualitatively, the spatial distribution and magnitude of the simulated fog are generally consistent with satellite and ground-based observations. Similarly, Lee et al. (2016) evaluated fog distribution simulations against satellite-derived cloud optical depth from satellite and concluded that the distributions of simulations and observations were generally comparable to each other."

[Figure]

Figure 4. (a, c) Distributions of ground-based fog observations (the circular points) and cloud optical depth from Himawari-8 products at 08:00 LST on 26 and 27 November 2018. (b, d) Simulated fog optical depth distributions in domain 03 at the corresponding time of observations. Time '2608LST' indicates 08:00 local standard time (LST) (LST = Universal Time Coordinated + 8 h) on 26 November 2018. The other time expressions follow the same logic.

**Minor Comments**

1. In the abstract the authors say "AFIs in the first fog…result in higher [droplet] number concentration …in Fog 2 than in Fog 1. For this to be true, my first thought was that AFIs in the first fog would have to reduce scavenging of aerosol and result in higher aerosol concentration in the second fog than would have been the case if the first fog hadn't formed. The authors don't show this. They do show that Fog 1 changes meteorological conditions, which might indirectly affect droplet concentration in Fog 2 by changing LWC in Fog 2, but starting the list at line 21 with droplet concentration rather than LWC implies (to me at least) that the main mechanism is an aerosol one: aerosol-fog interaction in Fog 1 affect aerosols in Fog 2, which then change droplet concentration in Fog 2, which then changes LWC and lifetime (the classic ACI pathway). The authors don't have any evidence for that (the mechanism is meteorology, not aerosols).

**Reply**: Sorry for the misleading sentences. We agree with the reviewer that the mechanism is meteorology, not aerosols. Therefore, we have revised the abstract (Page 1, Lines 21-25): "Our simulations indicate that the PBL conditions conducive to Fog2 formation are affected by ACI with high aerosol loading in Fog1; subsequently, PBL promotes ACI in Fog2, resulting in a higher liquid water content, higher droplet number concentration, smaller droplet size, larger fog optical depth, wider fog distribution, and longer fog lifetime in Fog2 than in Fog1."

2. Line 45 "proven" – I would say "showed" – a 'pivotal role' is not a mathematical concept so it is not really 'proved'.

**Reply**: We have revised the sentence accordingly (Page 3, Line 52): "The critical roles of aerosols and the planetary boundary layer (PBL) in these processes have been shown (Boutle et al., 2018; Niu et al., 2011; Quan et al., 2021). "

3.    Line 70 – what is the 'critical turbulence coefficient'? The reader should not need to look up the literature unless they are very unfamiliar with fog.

**Reply**: "The critical turbulence coefficient was the turbulence threshold for diagnosing whether turbulence suppressed fog or not. When the turbulence intensity within the fog did not exceed the critical turbulence coefficient, the fog persisted; however, when it surpassed its threshold, the fog dissipated (Zhou and Ferrier, 2008)." The above description is added (Page 4, Lines 83-86).

4.    Line 115   -the innermost simulation still has quite coarse spatial resolution. How well can this resolve the turbulence? Is there a sub-grid cloud parameterization in the model, or does the Grell 3D cumulus scheme lead to sub-grid variability in fog?

**Reply**: Turbulence is parameterised in the planetary boundary layer scheme. Based on closure theory, turbulent fluxes are calculated from gradients and parameterised vertical mixing coefficients. Besides, the parameterised vertical mixing coefficient also affects the vertical distribution of meteorological elements by the heat diffusion equation. Considering the consumption of computing cost, this kind of planetary boundary layer scheme is widely used in mesoscale numerical models (such as WRF), though its accuracy is not as good as the large eddy simulation.

There is a sub-grid cloud parameterisation in the MYNN2.5 planetary boundary layer scheme, instead of the Morrision microphysics scheme or the Grell 3D cumulus scheme. The sub-grid cloud parameterisation is found in the reference paper (Chaboureau and Bechtold, 2002), which is consistent with the source code in MYNN2.5 planetary boundary layer scheme. The sub-grid cloud water content is derived from a function of the normalized saturation deficit. So, sub-grid cloud parameterisation is considered in our paper.

We revised the sentences (Page 6, Line 147-148): "Turbulence is parameterised in the MYNN2.5 scheme and there is also a sub-grid cloud parameterisation (Chaboureau and Bechtold, 2002) in the MYNN2.5 scheme."

5. Line 165- what would be a perfect HSS score? Is the score calculated using each gridbox as input? Please be clearer about how this evaluation was done.

**Reply**: "A perfect HSS score is 1.0, indicating that simulations are identical to observations. We used the fog occurrence at the observation stations and the closest model grids as input for HSS score. In our study, the HSS score are 0.34 and 0.36 in Fog1 and Fog2, respectively, which are close to previous reports (Mecikalski et al., 2008; Xu et al., 2020; Yamane et al., 2010)." The above description is added in the revised manuscript (Page 9, Lines 202-205).

6. Figure 9: Is 'fog optical depth per unit height' the same as "average extinction coefficient through the fog"? It might help the reader to explain this in the caption.

**Reply**: Yes, 'fog optical depth per unit height' is the same as "average extinction coefficient through the fog". We have revised the sentences in the revised manuscript (Page 45, lines 887-888; Page 15, Lines 337 and 344).

**References**

Boutle, I., Price, J., Kudzotsa, I., Kokkola, H., and Romakkaniemi, S.: Aerosol–fog interaction and the transition to well-mixed radiation fog, Atmos. Chem. Phys., 18, 7827-7840, https://doi.org/10.5194/acp-18-7827-2018, 2018.

Chaboureau, J.-P. and Bechtold, P.: A Simple Cloud Parameterization Derived from Cloud Resolving Model Data: Diagnostic and Prognostic Applications, J. Atmos. Sci., 59, 2362-2372, https://doi.org/10.1175/1520-0469(2002)059<2362:ascpdf>2.0.co;2, 2002.

Lee, H.-H., Chen, S.-H., Kleeman, M. J., Zhang, H., DeNero, S. P., and Joe, D. K.: Implementation of warm-cloud processes in a source-oriented WRF/Chem model to study the effect of aerosol mixing state on fog formation in the Central Valley of California, Atmos. Chem. Phys., 16, 8353-8374, https://doi.org/10.5194/acp-16-8353-2016, 2016.

Mecikalski, J. R., Bedka, K. M., Paech, S. J., and Litten, L. A.: A Statistical Evaluation of GOES Cloud-Top Properties for Nowcasting Convective Initiation, Mon. Weather Rev., 136, 4899-4914, https://doi.org/10.1175/2008mwr2352.1, 2008.

Niu, S. J., Liu, D. Y., Zhao, L. J., Lu, C. S., Lü, J. J., and Yang, J.: Summary of a 4-Year Fog Field Study in Northern Nanjing, Part 2: Fog Microphysics, Pure Appl. Geophys., 169, 1137-1155, https://doi.org/10.1007/s00024-011-0344-9, 2011.

Quan, J., Liu, Y., Jia, X., Liu, L., Dou, Y., Xin, J., and Seinfeld, J. H.: Anthropogenic aerosols prolong fog lifetime in China, Environ. Res. Lett., 16, 044048, https://doi.org/10.1088/1748-9326/abef32, 2021.

Xu, X., Lu, C., Liu, Y., Gao, W., Wang, Y., Cheng, Y., Luo, S., and Van Weverberg, K.: Effects of Cloud Liquid-Phase Microphysical Processes in Mixed-Phase Cumuli Over the Tibetan Plateau, J. Geophys. Res.: Atmos., 125, https://doi.org/10.1029/2020jd033371, 2020.

Yamane, Y., Hayashi, T., Dewan, A. M., and Akter, F.: Severe local convective storms in Bangladesh: Part II, Atmos. Res., 95, 407-418, https://doi.org/10.1016/j.atmosres.2009.11.003, 2010.

Yan, S., Zhu, B., Zhu, T., Shi, C., Liu, D., Kang, H., Lu, W., and Lu, C.: The Effect of Aerosols on Fog Lifetime: Observational Evidence and Model Simulations, Geophys. Res. Lett., 48, https://doi.org/10.1029/2020gl091156, 2021.

Zhou, B. and Ferrier, B. S.: Asymptotic Analysis of Equilibrium in Radiation Fog, J. Appl. Meteorol. Clim., 47, 1704-1722, https://doi.org/10.1175/2007jamc1685.1, 2008.

---

## Author Response (AR2)

**Response to Referee #1 in Report #2**

Dear Referee,

Thank you for your positive and constructive comments. We have addressed your comments and the corresponding replies are listed below.

With regards,

Naifu Shao, Chunsong Lu*, and co-authors.

1) Validation. I still find it hard to understand how the evaluation was performed. There are now additional explanations of the evaluations measures, like the Heidke skill score. But the crucial information regarding the parameters used for calculating this and other scores does not seem to be present. This is crucial.

**Reply**: Thank you for your valuable comments. The crucial information regarding the parameters used for calculating Heidke skill score is revised (Page 9, Lines 201-208): "Elements $a$–$d$ are determined by the occurrence of fog at observation stations located in domain 03 and the closest model grids to those observations, as shown in Table 3. If fog events are both observed at stations and simulated at the closest model grids, we recognize those as "hits" and $a$ in Eq.1 represents the total number of "hits" during the entire fog event. Similarly, $d$ represents the number of "correct negatives" for the correct non-event simulations. On the other hand, if fog events are simulated but not observed, we recognize those as "false alarms" and $b$ represents the total number of "false alarms" during the entire fog event. Conversely, $c$ represents the total number of "misses", which indicates that fog events are observed but not simulated."

In addition to the Heidke skill score, information regarding the parameters used to calculate the other scores is given below.

The equations for root-mean-square error (RMSE) and mean bias (MB) are

$$\text{RMSE} = \sqrt{\sum_{i=1}^{n} \frac{(M_i - O_i)^2}{n}} \,, \tag{S1}$$

and

$$\text{MB} = \frac{1}{n}\sum_{i=1}^{n}(M_i - O_i)\,, \tag{S2}$$

where $M$ and $O$ represent the results from simulation and observation; $n$ is the total number of observation stations. They are added in the supplement. We also revised the main text (Page 39, Line 865): "The equations for RMSE and MB (Eq. S1-S2) are given in the supplement." The equations for the normalized mean bias (NMB), normalized mean error (NME), mean fractional bias (MFB), and mean fractional error (MFE) have already been given in the supplement (Eq. S3-S6). We have emphasized this information in the main text (Page 8, Line 186): "Eqs. S3–S6 in the supplement".

The paragraph starting at line 181 very vaguely introduces the evaluation, but (a) does not state explicitly which HIMAWARI product(s?) are being used

**Reply**: The Himawari product is level 2 full-disk cloud property data. We have added this information (Page 7, Line 154).

(b) what ground-based observations and of what parameter are being used

**Reply**: Ground-based observations refer to observations at meteorological stations. Observed visibility and relative humidity at those stations are used to evaluate fog distribution in Fig. 4. We have added the description (Page 8, Line191-193): "To identify observed fog at ground-based stations (the black circles in Fig. 4), we apply two criteria: visibility less than 1 km and relative humidity greater than 90% (Yan et al., 2020)."

(c) what time frames are covered

**Reply**: The time frames of Himawari products we used are 08:00 local standard time (LST) on 26 and 27 November 2018, respectively (Page 8, Lines 189-191).

(d) what resolutions are used

**Reply**: Spatial resolution of Himawari cloud product is 0.05°×0.05°. We have added this information (Page 7, Lines 159-160).

(e)what pre-processing steps, if any, were performed on the datasets

**Reply**: There are no pre-processing steps in using Himawari products.

(f) what are the references associated with these datasets

(g) what is known about product quality

**Reply**: We would like to reply to the two comments together. The references associated with these datasets are Bessho et al. (2016), Iwabuchi et al. (2018) and Yang et al. (2020). Himawari cloud products have been evaluated against the Moderate Resolution Imaging Spectroradiometer (MODIS) (Bessho et al., 2016; Letu et al., 2020) and cloud profiles from aircraft measurements (Zhao et al., 2020). Therefore, the quality of the Himawari cloud product is reliable. These information are added (Page 7, Lines 154-159).

The wording in the validation section is extremely vague. What is fog "magnitude"? (184), what is "generally consistent qualitatively"? (184). Please be specific.

**Reply**: We mean that the word "magnitude" refers to the value of fog optical depth, and the phrase "generally consistent qualitatively" indicates that the fog distribution and the value of fog optical depth show similarities between the simulation and observation. We have revised the sentence (Page 8, Lines 193-195): "Qualitatively, the value of fog optical depth and the fog spatial distribution in the simulation are roughly similar to

those observed by the Himawari satellite and at ground-based stations."

2) Representativity. While the state of the art chapter states a problem with a very wide scope, the study itself is focused on one situation in one particular location. In lines 444 onwards the authors present arguments for general applicability of the study, which however I find unconvincing. For a generalization I deem it necessary to state the range of conditions for which you believe your insights to hold. E.g., are the findings of this study transferrable to other fog situations in the same location? Other locations? Where, what conditions? Please specify and substantiate.

Reply: Thank you for your valuable comments. We have revised the sentences (Page 20, Lines 469-473): "This study focuses on a two-day radiation fog event in the Yangtze River Delta, China, which has a large population. The conclusions are expected to be applicable to radiation fog events in this region and other regions with similar human activities. It would be interesting to see if similar conclusions can be found in other fog types (e.g., advection fog) in other regions (e.g., ocean)."

**References**

Bessho, K., Date, K., Hayashi, M., Ikeda, A., Imai, T., Inoue, H., Kumagai, Y., Miyakawa, T., Murata, H., and Ohno, T.: An introduction to Himawari-8/9—Japan's new-generation geostationary meteorological satellites, Journal of the Meteorological Society of Japan. Ser. II, 94, 151-183, https://doi.org/10.2151/jmsj.2016-009, 2016.

Iwabuchi, H., Putri, N. S., Saito, M., Tokoro, Y., Sekiguchi, M., Yang, P., and Baum, B. A.: Cloud property retrieval from multiband infrared measurements by Himawari-8, Journal of the Meteorological Society of Japan. Ser. II, https://doi.org/10.2151/jmsj.2018-001, 2018.

Letu, H., Yang, K., Nakajima, T. Y., Ishimoto, H., Nagao, T. M., Riedi, J., Baran, A. J.,

Ma, R., Wang, T., and Shang, H.: High-resolution retrieval of cloud microphysical properties and surface solar radiation using Himawari-8/AHI next-generation geostationary satellite, Remote Sens. Environ., 239, 111583, https://doi.org/10.1016/j.rse.2019.111583, 2020.

Yang, Y., Zhao, C., and Fan, H.: Spatiotemporal distributions of cloud properties over China based on Himawari-8 advanced Himawari imager data, Atmos. Res., 240, 104927, https://doi.org/10.1016/j.atmosres.2020.104927, 2020.

Zhao, L., Zhao, C., Wang, Y., Wang, Y., and Yang, Y.: Evaluation of cloud microphysical properties derived from MODIS and Himawari-8 using in situ aircraft measurements over the Southern Ocean, Earth Space Sci., 7, e2020EA001137, https://doi.org/10.1029/2020EA001137, 2020.

**Response to Referee #2 in Report #1**

Dear Referee,

Thank you for your positive and constructive comments. We have addressed your comments and the corresponding replies are listed below.

With regards,

Naifu Shao, Chunsong Lu*, and co-authors.

The authors have responded well to my previous comments and those of the other reviewer, and the manuscript is much improved. The third simulation is useful and the new title and the additional clarity around the aerosol-fog interactions are clear improvements. The use of optical depth in Figure 4 is helpful. The suggestion that aerosol-fog interactions in the first fog affect the second for are now more convincing.

**Reply**: Thank you for your valuable comments.

However, I did not find clear responses of the authors, or changes in the manuscript, to address the comments of the editor, which are also very important. Perhaps the authors did not yet have the opportunity to respond. Therefore I recommend the manuscript be returned for minor revisions in order for the authors to respond to these additional suggestions (mainly around effects of radiative heating and cooling, expanding on lines 430-437, and written English). Specific discussion of the dependence of heating and cooling rates on droplet concentration is still lacking, and generally the written English can still be improved, especially in the new text, in line with the editor's suggestions.

**Reply**: Yes, we did not yet have the opportunity to respond. We now address the editor's and your comments together. The editor requested a discussion of the results in Petters et al. (2012), Fig. 1, and also suggested referring to a paper by Prabhakaran et al. (2023)

for further discussion. We have analysed the long-wave cooling and short-wave heating in the fog layer (Fig. 9b), referring to Fig. 1 in Petters et al. (2012) and Fig. 3h in Prabhakaran et al. (2023) in the main text. We have added sentences (Page 14, Lines 335-339): "When LWP is less than 20 g m$^{-2}$, vertically integrated long-wave cooling and short-wave heating are stronger under polluted conditions than those under clean conditions (Fig. 9b). This is similar to results from Petters et al. (2012) and Prabhakaran et al. (2023). Because $N_d$ shows a similar trend with LWP (Fig. S5), the dependence of heating and cooling rates on droplet concentration is consistent with the results based on LWP." We have also expanded the conclusion as you suggested (Page 20, Lines 457-459): "Radiative cooling and heating within the fog layer depend on LWP and $N_d$. When LWP in fog is less than 20 g m$^{-2}$, and higher aerosol loading enhances vertically integrated cooling and heating in optically thin fog."

[Figure]

**Figure 9.** (a) The timeseries of liquid water path (LWP) under polluted and clean conditions. The length of the bar represents standard deviation. (b) Dependence of fog-integrated radiative cooling or heating with LWP under polluted and clean conditions. $\theta_{LW}$ and $\theta_{SW}$ represent vertically integrated heating rate of potential temperature ($\theta$) within the fog layer due to long-wave radiation and short-wave radiation, respectively. Time '2512' indicates 12:00 local standard time (LST) (LST = Universal Time Coordinated + 8 h) on 25 November 2018. The other time expressions follow the same logic.

[Figure]

**Figure S5.** The timeseries of average liquid water path (LWP) in domain 03 under polluted and clean conditions (the red line: polluted conditions, the blue line: clean conditions). Time '2520' indicates 20:00 local standard time (LST) (LST = Universal Time Coordinated + 8 h) on 25 November 2018. The other time expressions follow the same logic.

The written English has been improved. Wiley Editing Services (https://editingse rvices. wiley.cn/) provides thorough English language editing.

More specific editorial comments:

(a) Abstract lines 17-18, 22,37: "PBL"->"the PBL", as is done correctly at line 75

**Reply**: We have added "the" as you suggested (Page 1-2, Lines 17-18, 22, 38).

(b) Sentence at line 163 needs improving.

**Reply**: We have revised the sentence (Page 7, Lines 170-171): "According to Fog1 dissipation time, clean conditions change to polluted conditions at 12:00 LST on 26 November 2018."

(c) L305 rendering->"leading to"

**Reply**: We have revised the phrase as you suggested (Page 14, Lines 320).

(d) The new additions starting at line 444 are repetitive, sound too defensive, and are too vague. I suggest this text be removed.

**Reply**: We have revised the sentences to show representativity as another referee suggested (Page 20, Lines 469-473): "This study focuses on a two-day radiation fog event in the Yangtze River Delta, China, which has a large population. The conclusions are expected to be applicable to radiation fog events in this region and other regions with similar human activities. It would be interesting to see if similar conclusions can be found in other fog types (e.g., advection fog) in other regions (e.g., ocean)."

**References**

Petters, J. L., Harrington, J. Y., and Clothiaux, E. E.: Radiative–dynamical feedbacks in low liquid water path stratiform clouds, J. Atmos. Sci., 69, 1498-1512, https://doi.org/10.1175/JAS-D-11-0169.1, 2012.

Prabhakaran, P., Hoffmann, F., and Feingold, G.: Evaluation of Pulse Aerosol Forcing on Marine Stratocumulus Clouds in the Context of Marine Cloud Brightening, J. Atmos. Sci., 80, 1585-1604, https://doi.org/10.1175/JAS-D-22-0207.1, 2023.

Dear Prof. Graham Feingold,

Thank you for your positive and constructive comments. We have addressed all the comments and the corresponding replies are listed below. Furthermore, the Wiley Editing Services (https://editingservices.wiley.cn/) provides thorough English language editing.

With regards,

Naifu Shao, Chunsong Lu*, and co-authors

**Response to Comments on July 3:**

Regarding Reviewer 1: As noted, the reviewer (and I) would like to see responses to all my (the Editor's) comments in general, and particularly regarding the effect of aerosol on LW and SW in optically thin fog and the quantification of LWP. Here I would like to see discussion of results in Petters et al. (2012), Fig, 1. You will see that for clouds with LWP < 20 g/m2, high aerosol concentration increases radiative cooling for the same LWP. In addition, in the SW, high aerosol concentration increases absorption (at the same LWP). You can also see further discussion in a paper by Prabhakaran et al. (2023) DOI 10.1175/JAS-D-22-0207.1, Pg 1590. Full engagement with ideas in these papers will enhance the impact of your work. My request for quantification of LWP becomes clear when you consider aerosol effects on cooling/heating at low LWP (Petters, Fig. 1).

**Reply**: We have analysed the quantification of LWP, especially the effect of aerosol on long-wave and short-wave radiation in optically thin fog (Fig. 9 and Fig. S5). We have added related description (Page 14, Lines 332-339): "As shown in Fig. 9a, LWP is larger under polluted conditions than that under clean conditions, particularly for Fog2. The average LWP in Fog1 and Fog2 under polluted conditions are 11.6 and 24.3 g m$^{-2}$, respectively. When LWP is less than 20 g m$^{-2}$, vertically integrated long-wave cooling and short-wave heating are stronger under polluted conditions than those under clean conditions (Fig. 9b). This is similar to the results from Petters et al. (2012) and

Prabhakaran et al. (2023). Because $N_d$ shows a similar trend with LWP (Fig. S5), the dependence of heating and cooling rates on droplet concentration is consistent with the results based on LWP." We have also expanded the conclusion according to reviewer's suggestion (Page 20, Lines 457-459): "Radiative cooling and heating within the fog layer depend on LWP and $N_d$. When LWP in fog is less than 20 g m$^{-2}$, higher aerosol loading enhances vertically integrated cooling and heating in optically thin fog."

[Figure]

**Figure 9.** (a) The timeseries of liquid water path (LWP) under polluted and clean conditions. The length of the bar represents standard deviation. (b) Dependence of fog-integrated radiative cooling or heating with LWP under polluted and clean conditions. $\theta_{LW}$ and $\theta_{SW}$ represent vertically integrated heating rate of potential temperature ($\theta$) within the fog layer due to long-wave radiation and short-wave radiation, respectively. Time '2512' indicates 12:00 local standard time (LST) (LST = Universal Time Coordinated + 8 h) on 25 November 2018. The other time expressions follow the same logic.

[Figure]

**Figure S5.** The timeseries of average liquid water path (LWP) in domain 03 under polluted and clean conditions (the red line: polluted conditions, the blue line: clean conditions). Time '2520' indicates 20:00 local standard time (LST) (LST = Universal Time Coordinated + 8 h) on 25 November 2018. The other time expressions follow the same logic.

Regarding Reviewer 2: Please address both points ('Evaluation' and Representativity), although don't use the word 'validation' since one cannot validate a model - one can only test or evaluate a model, which is in fact the spirit of the reviewer's comment.

**(1) Evaluation**

**Reply:** We have revised 'validation' to 'evaluation' (Page 7, Line153; Page 8, Line174). We have revised the sentences about fog distribution evaluation (Page 8, Lines 191-195): "To identify observed fog at ground-based stations (the black circles in Fig. 4), we apply two criteria: visibility less than 1 km and relative humidity greater than 90% (Yan et al., 2020). Qualitatively, the value of fog optical depth and the fog spatial distribution in the simulation are roughly similar to those observed by the Himawari satellite and at ground-based stations."

We have also revised the sentences about the calculation of the Heidke skill score (HSS) (Page 9, Lines 201-208): "Elements *a–d* are determined by the occurrence of fog at observation stations located in domain 03 and the closest model grids to those observations, as shown in Table 3. If fog events are both observed at stations and simulated at the closest model grids, we recognize those as "hits" and *a* in Eq.1 represents the total number of "hits" during the entire fog event. Similarly, *d* represents the number of "correct negatives" for the correct non-event simulations. On the other hand, if fog events are simulated but not observed, we recognize those as "false alarms" and *b* represents the total number of "false alarms" during the entire fog event. Conversely, *c* represents the total number of "misses", which indicates that fog events are observed but not simulated."

**(2) Representativity**

We have revised the sentences (Page 20, Lines 469-473): "This study focuses on a two-day radiation fog event in the Yangtze River Delta, China, which has a large population. The conclusions are expected to be applicable to radiation fog events in this region and other regions with similar human activities. It would be interesting to see if similar conclusions can be found in other fog types (e.g., advection fog) in other regions (e.g., ocean)."

**References**

Petters, J. L., Harrington, J. Y., and Clothiaux, E. E.: Radiative–dynamical feedbacks in low liquid water path stratiform clouds, J. Atmos. Sci., 69, 1498-1512, https://doi.org/10.1175/JAS-D-11-0169.1, 2012.

Prabhakaran, P., Hoffmann, F., and Feingold, G.: Evaluation of Pulse Aerosol Forcing on Marine Stratocumulus Clouds in the Context of Marine Cloud Brightening, J. Atmos. Sci., 80, 1585-1604, https://doi.org/10.1175/JAS-D-22-0207.1, 2023.

**Response to Comments on April 29:**

**Comments:**

I note that nowhere in your manuscript do you discuss the role of shortwave radiation and its affect on fog lifetime. This is an important part of the discussion. With increasing liquid water path (LWP) and increasing drop concentration, SW heating increases. There are also longwave aerosol-related effects: at low LWP ($< \sim 25$ g/m2); an increase in drop concentration will increase radiative cooling. I also note that there is no quantitative information on LWP, only its response (e.g., Table 4). This is really important information if one is to understand the radiation interactions.

**Reply**: Thank you for your suggestion. The detailed response to this comment is given in the "Response to Comments on July 3". Furthermore, we add the following discussion (Page 12-13, Lines 288-292): "Compared with clean conditions, the larger $\tau_t$ (mainly due to larger $\tau_c$) and delayed fog dissipation in polluted conditions reduce short-wave radiation reaching the ground (from $-46$ W m$^{-2}$ to $-121$ W m$^{-2}$) during the Fog1 dissipation time. This leads to a decrease in $T_{2m}$ (from $-0.2$ °C to $-1$ °C) and PBLH (from $-42$ m to $-118$ m), which further prolongs fog duration (Fig. 7)."

**A linguistic revision:**

1)  What are "conducive PBL conditions"? I think you mean "conducive to fog formation". Please make changes throughout.

**Reply**: Yes, we mean "conducive to fog formation". We have modified the phrase to be: "PBL conditions conducive to Fog2 formation". We have revised the entire article.

2)  change 'scenario' to 'event'

**Reply**: All 'scenario' in our article have been revised to 'event'.

3)  remove all "the" before "EXPn"

**Reply**: All "the" before "EXPn" has been removed.

4) dissipation time (not dissipate time)

**Reply**: All "dissipate time" has been revised to "dissipation time".

5) Fog 1 occurs under clean conditions (not Fog1 161 is under the clean condition)

**Reply**: We have revised "is under the clean conditions" to "occurs under clean conditions".

6) "under clean and polluted conditions" (remove 'the')

**Reply**: It has been removed in our article.

7) What is 'more remarkable AFI'? Do you mean stronger AFI? Please be clear.

**Reply**: Yes, 'more remarkable AFI' means stronger AFI. We have revised it in our article.

8) "can enhance cooling" do you mean "enhances cooling"? Please use clear causal language if that's what you mean. There are many instances "can affect", "can indicate".

**Reply**: Yes, we mean that "enhances cooling". We have revised all the related phrases.

9) You have a tendency to create new acronyms like AFI, FOD, TOD, which makes the manuscript less readable to the broader audience. The use of symbols significantly alleviates this problem (e.g., \tau_f, \tau_t). Even AFI might be unnecessary given the familiar ACI. (You could simply point out that ACI in fog has its own particular questions). Also, why N_f when N_d (drop concentration) or N_c (cloud droplet concetration) are widely used. And \tau_c would be better than \tau_f. As it is you use other standard cloud-related acronyms such as LWP, LWC.

**Reply**: Thank you for your suggestions to make the manuscript more readable to the broader audience. We have revised the acronyms and symbols accordingly. AFI, FOD, TOD and $N_f$ have been revised to ACI, $\tau_c$, $\tau_t$ and $N_d$, respectively.